# FASTATTENTION: EXTEND FLASHATTENTION TO NPUS AND LOW-RESOURCE GPUS FOR EFFICIENT LLM INFERENCE

## ABSTRACT

FlashAttention series has been widely applied in the inference of large language models (LLMs). However, FlashAttention series only supports the high-level GPU architectures, e.g., Ampere and Hopper. At present, FlashAttention series is not easily transferrable to NPUs and low-resource GPUs. Moreover, FlashAttention series is inefficient for multi- NPUs or GPUs inference scenarios. In this work, we propose FastAttention which pioneers the adaptation of FlashAttention series for NPUs and low-resource GPUs to boost LLM inference efficiency. Specifically, we take Ascend NPUs and Volta-based GPUs as representatives for designing our FastAttention. We migrate FlashAttention series to Ascend NPUs by proposing a novel two-level tiling strategy for runtime speedup, tiling-mask strategy for memory saving and the tiling-AllReduce strategy for reducing communication overhead, respectively. Besides, we adapt FlashAttention for Volta-based GPUs by redesigning the operands layout in shared memory and introducing a simple yet effective CPU-GPU cooperative strategy for efficient memory utilization. On Ascend NPUs, our FastAttention can achieve a $10.7\times$ speedup compared to the standard attention implementation. Llama-7B within FastAttention reaches up to $5.16\times$ higher throughput than within the standard attention. On Volta architecture GPUs, FastAttention yields $1.43\times$ speedup compared to its equivalents in `xformers`. Pangu-38B within FastAttention brings $1.46\times$ end-to-end speedup using FasterTransformer. Coupled with the propose CPU-GPU cooperative strategy, FastAttention supports a maximal input length of 256K on 8 V100 GPUs. All the codes will be made available soon.

## 1 INTRODUCTION

Recent years have witnessed the impressive performance of transformer-based large language models (LLMs) (Zhao et al., 2023; Vaswani et al., 2017; Kaplan et al., 2020) in understanding and generative tasks (Ainslie et al., 2023; Viggiato & Bezemer, 2023; Peng et al., 2023; Beltagy et al., 2020). It's noteworthy that the runtime and memory requirements of Transformer-based LLMs scale quadratically with the input sequence length. Dao et al. (2022) proposed FlashAttention series algorithms to reduce the runtime requirements and decrease memory usage from quadratic to linear complexity with regard to sequence lengths, of which FlashAttention2/3 (Dao, 2023; Shah et al., 2024) is widely used in the domain (Aminabadi et al., 2022; Shoeybi et al., 2020; NVIDIA, 2023b; Wu et al., 2023).

However, the FlashAttention series is typically designed for resource-rich Graphics Processing Units (GPUs) featuring high-level architectures, e.g., FlashAttention2 for Ampere and FlashAttention3 for Hopper (Jia & Van Sandt, 2021; Choquette, 2022). We identify three limitations of FlashAttention: 1) For architectures with lower capabilities, e.g., Volta (Choquette et al., 2018), and **non-CUDA architectures**, e.g., the architectures of Neural network Processing Units (NPUs), the existing FlashAttention project is not applicable. Additionally, `xFormers` implements the memory-efficient attention for V100 GPUs with suboptimal computational efficiency (Lefaudeux et al., 2022; Rabe & Staats, 2021). Moreover, a dedicated version of FlashAttention has been developed to support AMD GPUs (AMD, 2024). 2) The FlashAttention series exhibits inefficiency in **distributed inference** across multiple devices, arising from the communication overhead incurred by the `AllReduce` operations. 3) Under the constraints of limited device memory, FlashAttention can not enable

inference with **ultra-long sequences** on a single node within multi-NPUs and multiple low-resource GPUs. The limitations are caused by the significant differences in architectures and instruction sets, which pose challenges to adapting the existing FlashAttention for NPUs and low-resource GPUs, as detailed in § 3. In particular, directly transferring the workflow of the FlashAttention series to NPUs is inefficient, which means the techniques used in FlashAttention, such as tiling and work partitioning, typically can only exploit partial capabilities of non-CUDA architectures. As shown in Table 2.

Given the numerous inference systems that rely on low-resource GPUs and economical NPUs, failing to deploy FlashAttention in these systems could have significant adverse impacts. To address the issues mentioned above, we propose FastAttention, a pioneering adaptation of FlashAttention series for NPUs and low-resource GPUs with more efficiency. Without loss of generality, we design FastAttention for Ascend NPUs, e.g., Ascend 910B, and Volta-based GPUs, e.g., V100, serving as examples of the FlashAttention extension for NPUs and low-resource GPUs. Our contributions are summarized as follows:

- On NPUs, We propose a generalizable two-level tiling strategy, tiling-mask strategy and tiling-AllReduce strategy to save memory and improve runtime speedup for the adaption of FlashAttention. **Remarkably, to the best of our knowledge, we are the first to map FlashAttention series on NPUs**.

- We provide the implementation of FlashAttention tailored for low-resource GPUs, alongside a fine-grained CPU-GPU cooperative strategy to scale up the maximum input sequence length.

- Experimental results demonstrate that FastAttention achieves a $10.7\times$ speedup over the standalone implementation and provides a $5.16\times$ higher throughput compared to not using it on Ascend NPUs. On Volta-based GPUs, FastAttention achieves up to $1.43\times$ speedup when compared to its equivalents in `xformers`, enabling a $1.46\times$ lower latency and supporting a maximal input length of 256K when using FasterTransformer on a single node.

## 2  RELATED WORK

**Large Transformers:** Large Transformers, characterized by their extensive parameters and layers, are primarily employed for complex tasks such as natural language processing (NLP) and computer vision (Voulodimos et al., 2018). In these models, particularly in LLMs like GPT (Achiam et al., 2023), the attention mechanism plays a pivotal role, consuming a significant portion of computational resources. Although models such as Vision Transformers (ViT) and Diffusion Transformers (Spolaore & Wacziarg, 2009; Yuan et al., 2021) also incorporate attention mechanisms, the proportion of computation dedicated to attention in these models is relatively small. Consequently, the FlashAttention series is more specifically tailored to large transformer models, where attention computations are more prominent.

**FlashAttention series algorithms:** FlashAttention employs tiling and recomputation to minimize the number of memory access between the on-chip SRAM (a.k.a shared memory) and high bandwidth memory (HBM). It introduces frequent data flow via SRAM between Tensor Core and Cube Core (Choquette et al., 2021; Markidis et al., 2018). FlashAttention2 further optimizes the workflow of FlashAttention, exhibiting better parallelism and work partitioning. Based on the characters of newer GPU architectures, such as Hopper and Blackwell, FlashAttention3 hides the the non-GEMM operations under asynchronous General Matrix Multiplication (GEMM) with asynchronous instructions to further improve performance. Appendix A provides a more detailed description. However, FlashAttention2/3 only supports the resource-rich GPUs, neglecting low-resource GPUs and powerful NPUs. What's more, FlashAttention series lacks the capability to reduce the memory occupied by *attention_mask* and decrease the communication overhead introduced by `AllReduce`.

**Ultra-long sequence inference:** Limited device memory poses a significant constraint, rendering FlashAttention series incapable of supporting inference with ultra-long sequences (e.g., 256K) on a single node. Notably, the *offloading* is generally coupled with attention optimization for efficient memory utilization. For instance, both FlexGen (Sheng et al., 2023) and DeepSpeed-Inference (Aminabadi et al., 2022) design a classical *offloading* strategies that schedules data among GPUs, CPUs, and disks but lacks fine-grained pipeline design.

## 3 NPUs AND LOW-RESOURCE GPUs

**Similarity between NPUs and GPUs:** Most of the NPUs, such as Ascend NPUs, Hanguang NPUs, and Cambricon-series NPUs (Liao et al., 2021; Jiao et al., 2020; Song et al., 2023), are designed for high throughput and energy efficiency (Chen et al., 2020; Han & Yoo, 2023; Ahn et al., 2022). Taking Ascend NPUs as a representative, as shown in Figure 1, Ascend NPUs share similar design principles with GPUs, such as AI Cores corresponding to SMs in GPUs, Vector units corresponding to Cuda Cores and Cube units corresponding to Tensor Cores. Specifically, Cube units handle matrix computations while Vector units manage element-wise computations.

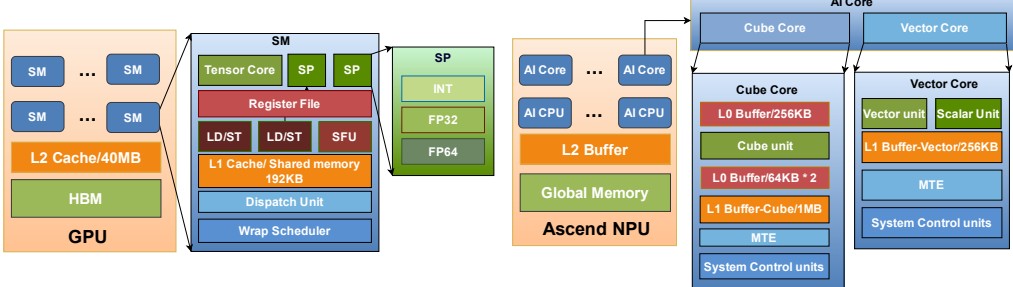

Figure 1: The comparison of architectures between resource-rich GPUs and Ascend NPUs

**Differences between NPUs and GPUs:** 1) **Decoupled architecture**. Each AI Core in Ascend NPUs integrates Cube, Vector, and Scalar units. The Cube units are decoupled from the Vector units, facilitating data exchange through the L2 buffer and global memory (GM), while Tensor Cores in GPUs interface with CUDA Cores via the shared memory. Consequently, the frequent data flow between Tensor Cores and CUDA Cores can limit the benefits of the decoupled architecture due to the synchronization overhead between the Cube and Vector units. In contrast, the decoupled architecture in Ascend NPUs enables seamless pipelining between Cube and Vector units, suggesting that the optimal programming model for Ascend NPUs follows a pipelined approach. This design allows element-wise computations by the Vector unit to overlap with matrix computations by the Cube unit, requiring a redesign of efficient attention mechanisms on NPUs from an overlapping perspective. 2) **Memory hierarchy.** In GPUs, Tensor Cores and Cuda Cores share access to the L2 cache, L1 cache, and register files. In contrast, the Cube units in NPUs are equipped with L0 buffers, which are absent in the Vector units, and feature a larger L1 buffer compared to the latter. The tiling method employed in FlashAttention fails to fully leverage the L1 buffer. To maximize the performance of NPUs, a meticulously designed data flow is essential. 3) **SDMA.** Ascend NPUs support System Direct Memory Access (SDMA) (Huawei, 2023b), which enables them to execute computation and communication in parallel. It's imperative to redesign FlashAttention algorithm to fully capitalize on SDMA, thereby reducing communication overhead and enhancing overall efficiency during inference.

**Low-resource GPUs versus high-end GPUs:** Low-resource GPUs exhibit similar architectures and memory hierarchies (HBM and SRAM) while different Tensor Cores for matrix computations compared to the resource-rich GPUs, resulting in distinct requirements for data layout in SRAM. This variation in data layout presents significant challenges when extending FlashAttention series to low-resource GPUs. Notably, high-level architectures like Ampere and Hopper already feature efficient attention implementations, i.e., FlashAttention series. GPUs with lower-level architectures than Volta do not possess Tensor Cores, which are essential for implementing various efficient attention mechanisms. For this reason, we take the Volta-based GPUs as representatives in our work.

**Why we refer to FastAttention as an extension of FlashAttention:** Due to the similarities between GPUs and NPUs, some basic ideas inside FlashAttention series, such as fused blocked GEMM and online softmax, can be also employed on NPUs with non-trivial implementation. However, significant differences in architecture, memory hierarchy, and SDMA necessitate a tailored redesign to fully leverage NPU capabilities. FlashAttention series is also not applicable for low-resource GPUs, as they require distinct data layouts and block partitioning strategies to utilize Tensor Cores. Moreover, FlashAttention is not optimized for multi-NPU or multi-low-resource GPU scenarios. In contrast, FastAttention incorporates only a few basic ideas from FlashAttention while introducing substantial novel techniques as detailed below, extending FlashAttention in both design techniques and applicability.

# 4 METHODOLOGY

We considers different application scenarios to design FastAttention: 1) In single-NPU scenarios, FastAttention features a two-level tiling strategy for the NPU's computational power utilization and an architecture-independent tiling-mask strategy for memory saving. 2) In multi-NPU scenarios, building upon the prior method, FastAttention further integrates the tiling-AllReduce strategy to minimize the communication overhead. 3) For low-resource GPUs such as those with Volta architecture, FastAttention applies the standard FlashAttention2 kernel and redesigns the shared memory layout of operands in FlashAttention2 to adapt the instructions of Volta architectures. 4) In case GPU memory is insufficient for inference, FastAttention equips a fine-grained CPU-GPU cooperative strategy with the prior standard kernel to fully utilize the CPU's computing power and memory.

## 4.1 FASTATTENTION ON A SINGLE NPU

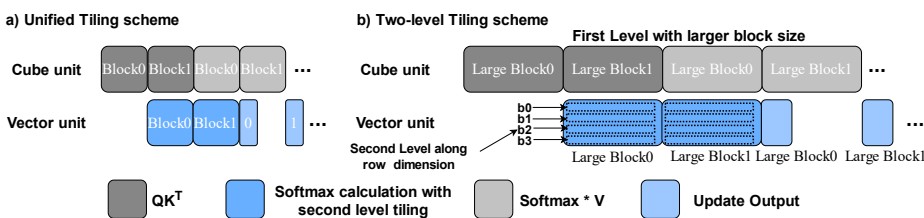

Figure 2: a) The unified tiling scheme with the fine-grained pipeline of Vector and Cube units; b) The two-level tiling strategy that employs the larger block size in the first level and maintains the smaller block size in the second level.

In this section, we delve into the redesign of the FlashAttention2 operator for Ascend NPUs. Initially, drawing upon the standard FlashAttention2 implementation for GPUs, we develop a standard FlashAttention2 kernel for Ascend NPUs, which employs the unified tiling strategy illustrated in the left of Figure 2. Specifically, considering the L1 buffer sizes in Ascend NPUs, we distribute the $Q$ matrix across the Ascend NPU's AI Core units and split the input matrices $K$ and $V$ along the sequence length (S dimension) into small blocks.

**Pipeline:** Each of these small tiling blocks follows computations sequentially executed by Cube and Vector units. This design allows Vector and Cube units to work in tandem, achieving a better pipeline for efficient parallel computation. For instance, when the block0 performs the $Exp$ calculation by the Vector unit, the block1 will perform the matrix multiplication of $Q * K^T$ by the Cube unit.

**NPU-specific two-level tiling strategy:** The unified tiling strategy employs small block size, leading to the frequent data flow between Cube unit and Vector unit via L2 buffer, which in turn introduces significant synchronization overhead. Additionally, the distinct computational characteristics and discrepancy in L1 buffer size between Cube and Vector units result in underutilization of the Cube's L1 buffer. To address the issues, we propose a novel two-level tiling strategy. The two-level tiling strategy is depicted in the right of Figure 2. In the first level, we adopt larger block sizes than the former implementation for Cube unit, which can effectively decrease the number of synchronizations. Additionally, we optimize the pipeline parallelism of Vector and Cube by utilizing the double-buffering technique on GM. Furthermore, this design allows the Cube unit to load larger continuous blocks for the utilization of memory bandwidth. Considering the limited L0 and L1 size, we split the large blocks into several small blocks along row dimension for Vector unit. This design provides a more refined pipeline over the multi-level memory of each computing unit. What's more, we also apply the double-buffering technique to overlap the data transfer and computation in the second level. The more detailed mathematical equations for the two-level tiling are provided in Appendix B.

**Tiling-mask:** Besides, we propose an architecture-agnostic tiling-mask strategy to eliminate the memory requirement for *attention_mask*. Specifically, we implement an attention_mask generator that uses a small mask matrix with the dimensions of $(2 * M) * (2 * M)$ ($M$ represents the maximal block size) to substitute the complete *attention_mask* matrix with the dimensions of $S * S$ ($S$ represents the sequence length). The complete *attention_mask* matrix is a lower triangular matrix. Due to the tiling strategy implemented in FlashAttention2, the corresponding *attention_score* blocks need to be

masked by a smaller mask matrix (abbreviated as *B-mask*) with the same block size dimension. It is important to note that the block size (abbreviated as $b$) of the B-mask should be less than $M$.

The attention_mask generator can generate the *B-mask* matrices required for any *attention_score* block by the mask matrix with dimensions of $(2 * M) * (2 * M)$ (abbreviated as *M-mask*). For instance, as depicted in Figure 3, a *M-mask* matrix $(6 * 6)$ can be split into multiple *B-masks* when $b$ is 3. Each *attention_score* block can search for the required *B-mask* within the *M-mask* matrix by employing mathematical transformations.

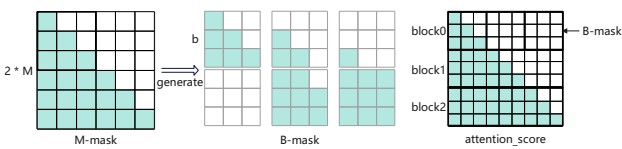

Figure 3: In case $b = 3, M = 3$, a *M-mask* matrix can be split into 6 *B-mask* matrices required by any given blocks through shifting.

Furthermore, there are two particular scenarios: all values within the *B-mask* are 0, and all values within the *B-mask* are 1. In the first scenario, we can directly skip the computation for that block, saving approximately 50% of the Cube computation. For the scenario where all values within the block are 1, we can directly skip the computation of $Q * K^T + mask$, thereby reducing the calculation for the vector. Tiling-mask can significantly reduce memory consumption. For instance, the *attention_mask* matrix requires 8GB GPU memory ($batchsize = 1, sequence\_length = 64K$) while *M-mask* ($M = 512$) only demands 256KB.

## 4.2 FASTATTENTION IN MULTI-NPU SCENARIOS

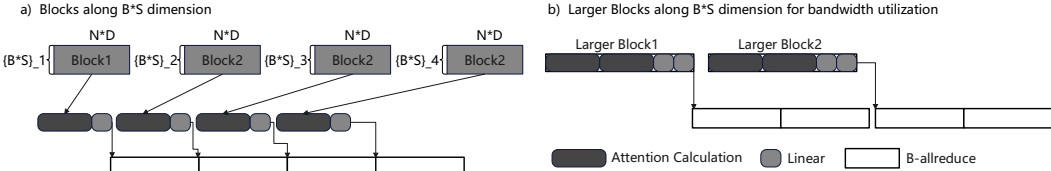

Figure 4: The pipeline of FastAttention with tiling-AllReduce strategy using different block sizes.

**Tiling-AllReduce strategy for multi-NPUs:** When each NPU completes the *attention* and *Linear* calculation in multi-NPU scenarios, `AllReduce` is employed to gather the computation results. Note that FlashAttention series accelerates the calculation of $x_o = Softmax(\frac{QK^T}{\sqrt{d}})V$, the Linear calculation implys the multiplication of $x_o$ and the $W_o$ matrix. Building upon the prior work, we fuse the *attention* and *Linear* calculation into a more efficient kernel and employ the tiling-AllReduce strategy to reduce the communication overhead. Specifically, we split the `AllReduce` operation into multiple `AllReduce` operations (abbreviated as `B-allreduce`) on a per-block basis. The `B-allreduce` operations are overlapped with block calculations to improve performance.

As shown in Figure 4, we partition the input matrix Q with shape $B * S * N * D$ (batch size, sequence length, number of heads, and head dimension defined by B, S, N, D, respectively) into multiple blocks along the dimension $B * S$ on each Ascend NPU. For each independent block, FastAttention will sequentially complete the *attention* calculation, *Linear* calculation, and the `B-allreduce` communication. The `B-allreduce` in a block can be overlapped with the calculation of other blocks due to the SDMA supported by Ascend NPUs. In this way, except for the first block, the other blocks can efficiently reduce the communication overhead. In order to minimize the impact of the computation time of the first block, we assign smaller computation tasks to the first block. Moreover, the tiling method minimizes the data transferred per communication, leading to underutilization of bandwidth capacity. Therefore, we enlarge the block size to achieve better bandwidth utilization. The new tiling method is illustrated in the right of Figure 4. The formalized description of this strategy can be found in the Appendix B.

## 4.3 FASTATTENTION ON LOW-RESOURCE GPUS

In this section, we provide a computation-efficient adaptation of FlashAttention2 for Volta-based GPUs. The main issue encountered in porting FlashAtention2 to the Volta-based GPUs is the hard-coded use of MMA (matrix multiply and accumulate) operations, which is supported only in Nvidia

architectures above or equal to Ampere. The code assumes the use of the `m16n8k16` and `m16n8k8` Tensor Core instructions, while the V100 supports only `m8n8k4` one. As shown in Figure 5, Volta architecture implements an MMA instruction where a group of 8 threads called a quadpair (QP) collaborate to share data and perform an 8x8x4 MMA. In this figure, **T** stands for the index of the thread in a warp. Since a warp is 32 threads wide, it would perform an MMA across 4 QPs for a tile size of 16x16x4. While Ampere MMA instruction `m16n8k16` operates at the granularity of 1 Warp. Hence, these operations have completely different partitioning of the input data and the resulting output. Furthermore, the implementation of many FlashAttention2 methods, such as softmax, causal masking, and transposing the data layout, can only work for Ampere and above. For example, the function `convert_layout_acc_rowcol` that transforms the layout cannot be used for Volta MMA.

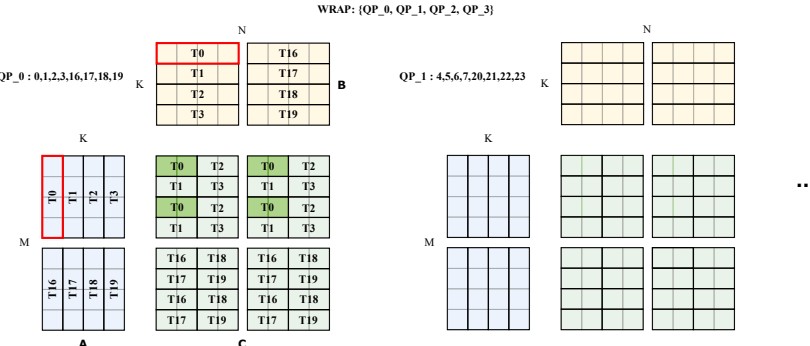

Figure 5: An example of MMA instruction `m8n8k4` for Volta.

To address the issues, we carefully redesign the data layout in SRAM with the CuTe library (Nvidia, 2023) to accommodate Volta instruction sets. And we base on Volta m8n8k4 instruction with FP16 accumulators to create a converter for the data layout redesign. Our codes are flexible and adaptable for any Volta-based GPU. The more detailed mechanism behind this data layout redesign can be found in Appendix C. The CuTe library typically focuses on new-generation GPUs and lacks the SRAM and HBM layouts examples for the Volta architecture. We delve extensively into the CuTe sources to redesign the data layout to eliminate bank conflicts (Nvidia, 2024) in SRAM access and make coalesced access to HBM. In the end, we successfully port the FlashAttention2 implementation on Volta-based GPUs, which provides better performance.

### 4.4 FASTATTENTION FOR ULTRA-LONG SEQUENCES IN MULTIPLE LOW-RESOURCE GPUS SCENARIOS

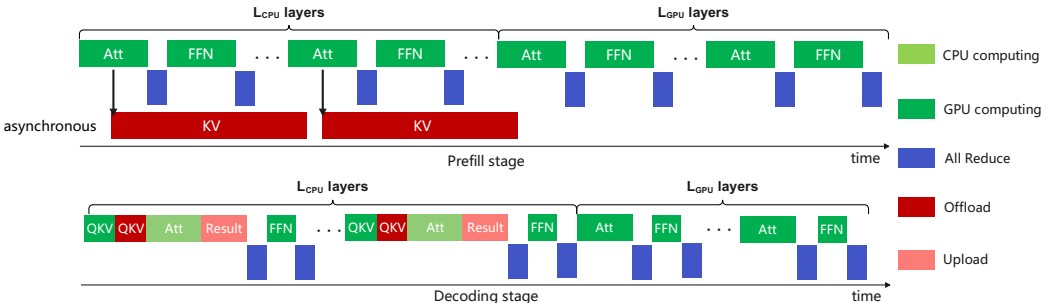

Figure 6: The pipeline of the CPU-GPU collaborative strategy during *prefill* stage and *decoding* stage.

For inference on multiple low-resource GPUs, we propose a **CPU-GPU cooperative strategy** coupled with our efficient attention kernel to extend the maximal input sequence length and exhibit better performance than the classical *offloading*. The detailed description of this strategy is as follows: 1) In case the GPU memory is sufficient for inference, there is no need to employ the *offloading* method. 2) Otherwise, our strategy will manage the memory of CPUs and GPUs. The strategy calculates the value of $L_{CPU}$ and $L_{GPU}$, which means the number of layers where the KV cache is stored on CPUs and the number of layers where the KV cache is stored on GPUs, respectively. The $L_{GPU}$ and $L_{CPU}$ can be computed by: $L_{GPU} = \frac{M_{GPU} - \frac{M_w}{n} - M_{mid} - M_{vocab}}{M_{kv}}$ and $L_{CPU} = L - L_{GPU}$. The $M_{GPU}$

refers to a single GPU memory. $M_w$, $M_{kv}$, $M_{vocab}$ and $M_{mid}$ represent the memory occupied by model weights, KV cache of one layer, the vocabulary matrix and the intermediate results on a single GPU, respectively. The total number of transformer layers is denoted as $L$, while the number of GPUs is $n$. For details, please see our Appendix D. 3) As shown in the top of Figure 6, during the *prefill* stage, the KV cache of the pre-$L_{CPU}$ layers will be asynchronously offloaded to the CPUs after the calculation of the KV matrix, which eliminates the offloading overhead. 4) During the *decoding* stage, as described in the bottom of Figure 6, our strategy offloads the QKV matrix of the pre$L_{CPU}$ and uses CPUs to finish the *attention* calculation. It utilizes multi-threading and vectorized instructions, e.g., AVX512, to reduce the calculation latency using CPUs. The calculation results will be uploaded to GPUs and finish the FNN calculation. For the rest of $L_{GPU}$ layers, all the calculations will be completed by GPUs.

## 5 PERFORMANCE EVALUATION

### 5.1 OVERVIEW

We conduct extensive evaluations on our FastAttention. We use the closed-source PanGu-series and open-source LLaMA-series models (Zeng et al., 2021; Ren et al., 2023; Touvron et al., 2023b; Huawei, 2023a) to demonstrate the superior performance and generalizability of FastAttention. Table 1 elaborates the model configurations. We conduct the experiments on two types of hardware: Ascend 910B NPUs and Nvidia Tesla V100 GPUs.

| Model name | # params (B) | # Layers | Heads | Head_dim | FFN size |
|---|---|---|---|---|---|
| PanGu-38B | 38 | 40 | 40 | 128 | 20480 |
| OPT-30B | 30 | 48 | 56 | 128 | 28672 |
| LLaMA2-7B | 7 | 32 | 32 | 128 | 11008 |
| LLaMA2-70B | 70 | 80 | 64 | 128 | 28672 |
| LLaMA-65B | 65 | 80 | 64 | 128 | 22016 |

Table 1: The model configurations used for the model inference performance evaluation.

First of all, we clarify the definition of standard attention: the naive implementation of matrix operations following the equation $Softmax\{\frac{\boldsymbol{QK}^T}{\sqrt{d}}\}\boldsymbol{V}$ without optimizations like operator fusion and online Softmax. These experiments show that FastAttention is 4.85-10.7$\times$ faster than the standard attention implementation on an Ascend NPU and up to 1.40$\times$ faster on 8 Ascend 910B. The system within FastAttention yields up to 5.1$\times$ throughput compared to the baseline on an Ascend 910B, while demonstrating comparable latency and throughput on 8 Ascend 910B. Moreover, FastAttention achieves a speedup of 1.43$\times$ than xformers' FlashAttention implementation and 1.48$\times$ over the classical *offloading* for ultra-long sequence inference on V100 GPUs. Furthermore, FastAttention extends the maximum input sequence length from 16K to 256K and reaches up to 1.46$\times$ speedup compared to a baseline without FastAttention on a machine equipped with 8 V100 GPUs. None of our optimizations compromise accuracy and FastAttention is orthogonal to techniques such as quantization. Additional experiment details are in Appendix E. Note that FlashAttention series is not applicable for Ascend NPUs and V100, making a direct comparison with FastAttention infeasible in these experiments.

### 5.2 THE OPERATOR-LEVEL PERFORMANCE EVALUATION OF FASTATTENTION

#### 5.2.1 FASTATTENTION ON A SINGLE ASCEND NPU

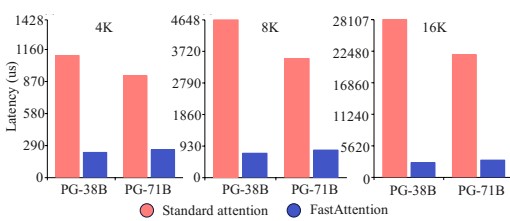
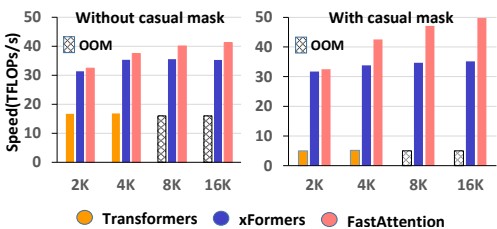

Figure 7: Performance comparison of FastAttention and standard attention with batch size 1 during the *prefill* stage on the Ascend 910B.

Figure 8: Performance comparison of FastAttention and xformers' FlashAttention with batch size 8 and hidden dimension 2048 during the *prefill* stage on a V100.

We compare the latency of the FastAttention operator during the *prefill* stage with that of the standard attention implementation. We use PanGu-38B and PanGu-71B to evaluate the optimization effects of the FastAttention operator on an Ascend 910B. All experiments use a batch size (B) of 1, with

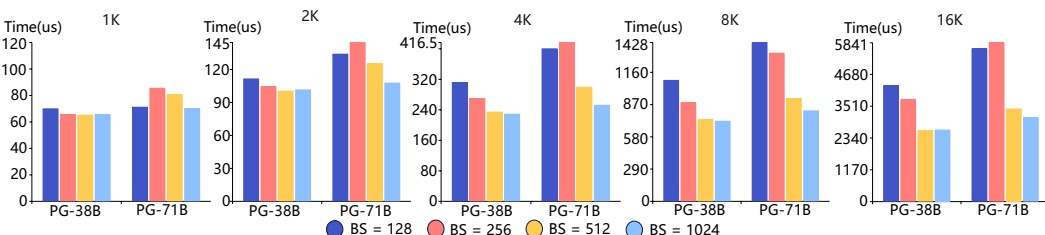

Figure 9: The latency comparison of FastAttention with different block sizes on an Ascend 910B across sequence lengths from 1K to 16K during the *prefill* stage.

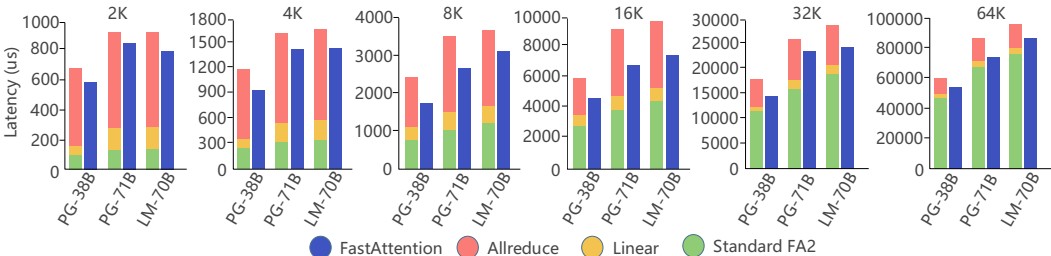

Figure 10: The performance of FastAttention on eight Ascend 910B NPUs with sequence length from 2K to 32K and batch size 1 during the *prefill* stage.

5 heads (N) and a head dimension (D) of 128 for PanGu-38B, and 4 heads and a head dimension of 128 for PanGu-71B. Figure 7 demonstrates the significant performance improvements using the FastAttention operator. Embedded in PanGu-38B and PanGu-71B, the FastAttention operator can achieve at most $10.7\times$ and $7.1\times$ speedup, respectively.

Furthermore, we conduct experiments to analyze the impact of the two-level tiling strategy on performance improvement. The experiments compare the latency of the FastAttention operator with different block sizes of the first level across multiple input sequence lengths on an Ascend 910B. The results for $BS = 128$ ($BS$ represents the basic block size) are treated as the baseline. Figure 9 shows that with the 4K input sequence length, the proposed strategy helps reduce latency by about 26% and 37% for PanGu-38B and PanGu-71B, respectively. Moreover, with the 8K and 16K input sequence length, the operator latency decreases by 33% and 38% for PanGu-38B, and the reductions are 43% and 45% for PanGu-71B, respectively. These results demonstrate the efficiency of our optimization strategy, especially for long sequence lengths.

### 5.2.2 FASTATTENTION ON MULTI-NPUS

Table 2: Ablation study of proposed strategies on NPUs.

| Operator | Tiling-mask | Unified tiling | Two-level tiling | Tiling-AllReduce | Speedup |
|---|---|---|---|---|---|
| Standard attention | X | X | X | X | 1 |
| FastAttention | ✓ | X | X | X | 1 |
| FastAttention | X | ✓ | X | X | 2.55-7× |
| FastAttention | X | X | ✓ | X | 3.65-10.7× |
| FastAttention | X | X | ✓ | ✓ | 4.23-15× |
| FastAttention | ✓ | X | ✓ | ✓ | 4.23-15× |

In these experiments, we compare the latency of FastAttention in terms of the total latency involved in the unfused FastAttention kernel (the implementation in § 4.1), *Linear* operator, and the `Allreduce` operation. We conduct evaluations of the FastAttention using PanGu-38B, PanGu-71B and LLaMA2-70B with varying sequence lengths on eight Ascend 910B NPUs. Batch size is 1 in all the experiments. Figure 10 demonstrates a significant speed improvement of our FastAttention. For PanGu-38B, our FastAttention achieves a speedup ranging from $1.16\times$ to $1.40\times$ across sequence lengths from 2K to 32K. For PanGu-71B, it can achieve a performance improvement of 7.4%, 12.3%, 24.2%, 26.1%, and 21.3% for sequence lengths of 2K, 4K, 8K, 16K, and 32K, respectively. FastAttention achieves up to $1.3\times$ lower latency for LLaMA2-70B. Notably, as the sequence length increases, the FastAttention operator typically can achieve more performance improvement due to the increased proportion of overlapping time. Furthermore, we give the ablation study of FastAttention for Ascend NPUs in Table 2 to demonstrate the effectiveness of the proposed strategies. Note that the tiling-AllReduce

strategy has to be built upon the two-level tiling strategy. Therefore, we don't test the tiling-AllReduce strategy independently. With the two-level strategy, our FastAttention reaches up to $10.7\times$ speedups and realizes a maximum speedup of $15\times$ coupled with tiling-AllReduce strategy. FastAttention also demonstrates significant performance improvements for small Transformers, such as Vision Transformers. The relevant experimental results are presented in Appendix E.

### 5.2.3 FASTATTENTION ON LOW-RESOURCE GPUs

We measure the runtime of the FastAttention operator across different sequence lengths and compare it to the FlashAttention operator in `xformers`. We compare the two operators on a single V100 GPU under two settings: without and with causal masks. Benchmark settings are as follows: 1) sequence length varies from 2k to 16k, 2) batch size is set to 8, 3) hidden dimension to 2048, and 4) head dimension to 64. To calculate the FLOPs, we used the formula $4 \cdot batchsize \cdot seqlen^2 \cdot head\ dimension \cdot number\ of\ heads$. Figure 8 demonstrates that FastAttention always exhibits higher TFLOPs/s compared to the counterpart in `xformers`. Without causal mask, our operator achieves speedup of $1.03\times$, $1.06\times$, $1.12\times$, and $1.17\times$ for sequence lengths of 2K, 4K, 8K, and 16K, respectively. With causal mask, as the sequence length increases, FastAttention can achieve a maximum speedup of $1.43\times$. Additionally, we compare FastAttention with the vanilla implementation in Transformers (huggingface, 2024),achieving maximum speedups of $8.2\times$ and $2.2\times$ with and without causal mask, respectively.

### 5.2.4 FASTATTENTION WITH ULTRA-LONG SEQUENCE ON MULTIPLE LOW-RESOURCE GPUs

Table 3: The performance comparison of our CPU-GPU strategy and classical *offloading* strategy.

| Seq_length | Classical *Offloading* | | | FastAttention | | | |
|---|---|---|---|---|---|---|---|
| | | | | $L_{CPU}$ layers | | | $L_{GPU}$ layers |
| | Upload(ms) | GPU_Calc(ms) | Total(ms) | CPU_Calc(ms) | Off_Upload(ms) | Total(ms) | GPU_Calc(ms) |
| 1K | - | 0.058 | 0.058 | - | - | - | 0.058 |
| 2K | - | 0.068 | 0.068 | - | - | - | 0.068 |
| 4K | - | 0.095 | 0.095 | - | - | - | 0.095 |
| 8K | - | 0.17 | 0.17 | - | - | - | 0.17 |
| **16K** | $3.58 \pm 0.43$ | 0.312 | 3.892 | 2.676 | 0.043 | **2.719** | 0.312 |
| **32K** | $6.98 \pm 0.46$ | 0.568 | 7.548 | 5.30 | 0.045 | **5.345** | 0.568 |
| **64K** | $13.13 \pm 0.53$ | 1.123 | 13.66 | 10.625 | 0.06 | **10.685** | 1.123 |
| **128K** | $25.61 \pm 0.4$ | 2.088 | 27.698 | 18.66 | 0.061 | **18.721** | 2.088 |
| **256K** | $50.81 \pm 0.39$ | 4.11 | 54.92 | 37.74 | 0.066 | **37.806** | 4.11 |

- represents that the system doesn't necessitate *offloading* strategies.
The data highlighted in the gray background section signifies the total latency required by *attention* calculation.

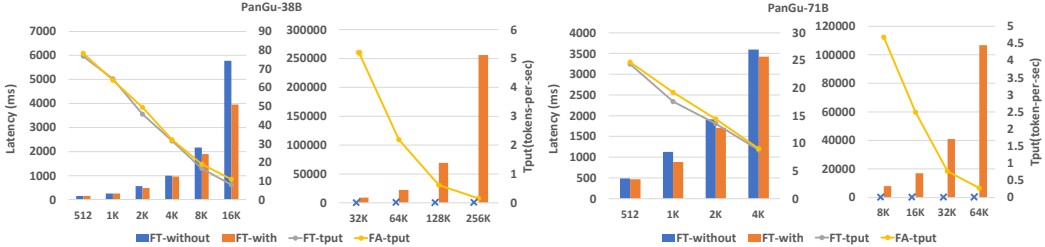

Figure 11: Latency and throughput comparison of FasterTransformer with and without FastAttention for different models and sequence lengths on eight V100 GPUs.

We measure the latency of *attention* calculation using FastAttention integrated with our CPU-GPU cooperative strategy, as well as only using the classical *offloading* strategy, respectively. The classical *offloading* offloads the KV cache from GPUs to CPUs and uploads the KV cache to GPUs when necessary. We use PanGu-38B to conduct the experiments with different sequence lengths and batch size 1 on eight V100 GPUs. Table 3 shows the latency breakdown to the *attention* calculation of one transformer layer. For the classical *offloading*, `Upload` implies the latency of uploading KV cache to a GPU. For the FastAttention, `CPU_Calc` time represents the latency of *attention* calculation using a CPU, `Off_Upload` contains the latency of offloading QKV matrix and that of uploading the results. Both `Total` means the total latency of the *attention* calculation, and `GPU_Calc` implies the latency of *attention* calculation using a GPU. Due to the same operators with different strategies applied, the values of `GPU_Calc` in `FastAttention` and `Classical Offloading` are similar.

In Table 3, it can be observed that the sequence length can reach up to 256K using our strategy on eight V100 GPUs. For the pre-$L_{CPU}$ layers, FastAttention using our strategy is 1.27-1.48× faster than using classical *offloading*. For the rest $L_{GPU}$ layers, it's up to 13.36× faster than using classical *offloading*. Specifically, it is evident that `Off_Upload` remains almost constant latency, as our strategy solely necessitates uploading results of fixed dimensions during the *decoding* phase. Employing our strategy, the `CPU_Calc` latency is notably lower than the `Upload` in classical offloading. This discrepancy arises from the PCIe for data transfer on V100, which provides a mere theoretical bidirectional bandwidth of 32GB/s. Moreover, the real-world bandwidth is often affected by various factors, which may prevent it from reaching the theoretical peak.

## 5.3 THE END-TO-END PERFORMANCE OF FASTATTENTION

| Seq_length | PanGu-38B | | PanGu-71B | |
|---|---|---|---|---|
| | Latency (ms) | token-per-sec | Latency (ms) | token-per-sec |
| 4K | 240.81 | 95 | 539.14 | 34 |
| 8K | 292.33 | 88 | 1052.49 | 33 |
| 32K | 1393.42 | 76 | 4948.33 | 25 |

Table 4: End-to-end performance evaluation of FastAttention on 8 Ascend 910B.

| Seq_length | OPT-30B | | LLaMA-65B | |
|---|---|---|---|---|
| | Latency (ms) | token-per-sec | Latency (ms) | token-per-sec |
| 512 | 270.5 ± 9.35 | 20.25 ± 0.7 | 513.15 ± 16.31 | 10.57 ± 0.33 |
| 1K | 384.74 ± 30 | 16.27±1.26 | 1046.79 ± 43 | 6.73 ± 0.27 |
| 2K | 691.67 ± 100 | 11.59 ± 1.67 | 2206.95 ± 200 | 4.08 ± 0.4 |
| 4K | N/A | N/A | 3848.61 ± 300 | 2.35 ± 0.18 |
| 8K | N/A | N/A | N/A | N/A |

N/A means that the sequence lengths surpass the model limitation or encounter some system errors during the experiments.

Table 5: End-to-end performance evaluation of Deepspeed on 8 V100.

We use two performance metrics: (i) latency, i.e., end-to-end time to generate one token for an input sequence, and (ii) token throughput, i.e., tokens-per-second processed. We measure the latency of generating one token with an input sequence of different lengths, which reflects the high computational efficiency of FastAttention. For throughput, we measure the performance with an input sequence of varying lengths while generating 50 tokens at a time.

Firstly, we measure the throughput with an input prompt of 512 tokens using LLaMa2-7B on an Ascend 910B, thereby demonstrating the performance of FastAttention on a single NPU. As shown in Table 6, the system within FastAttention achieves up to 5.16× higher throughput. Then, we conduct the experiments with PanGu-38B and PanGu-71B on eight Ascend 910B NPUs. Table 4 demonstrates the excellent performance of FastAttention on 8 Ascend 910B. For an input sequence of varying lengths, it consistently demonstrates low latency and high throughput.

| Batch_size | Throughput (token-per-sec) | |
|---|---|---|
| | Standard attention | FastAttention |
| 1 | 11.03 | 56.974 |
| 8 | 91.61 | 436.1 |
| 16 | 158.34 | 746.27 |

Table 6: The throughput comparison within and without FastAttention using PyTorch for different batch sizes on an Ascend 910B.

Moreover, we measure the latency and throughput of PanGu-38B and PanGu-71B on eight V100 GPUs. Note that we evaluate the performance of FasterTransformer (FT) (NVIDIA, 2023a) with and without FastAttention, respectively. Figure 11 demonstrates the effectiveness of our optimization strategies. FT without FastAttention can only support sequences up to 16K in length, while it can handle up to 256K using FastAttention. This is attributed to the fine-grained CPU-GPU cooperative strategy. What's more, for PanGu-38B, FT with FastAttention achieves a speedup of up to 1.46× over FT without FastAttention while 1.28× for PanGu-71B. FastAttention enables FT to surpass the GPU memory limitations and achieve superior performance. We also conduct experiments with OPT-30B (Zhang et al., 2022) and LLaMA-65B (Touvron et al., 2023a) using Deepspeed on eight V100 GPUs. As shown in Table 5, the torch-version DeepSpeed exhibits lower inference performance compared to FT. We analyze that torch-version Deepspeed doesn't utilize asynchronous methods, such as CUDA graphs, introducing additional invocation overheads and driver overheads. Therefore, DeepSpeed was not selected for comparative experiments.

## 6 CONCLUSION

In this paper, we propose FastAttention, an extension of FlashAttention2 for both NPUs and low-resource GPUs, enabling longer input sequence lengths and lower inference latency. FastAttention contains a series of insightful strategies and optimizations that are theoretically generalizable to all of the NPUs and low-resource GPUs with similar architectures. Extensive experiments have demonstrated the excellent efficiency and generalization of FastAttention on multiple LLMs. In future work, we will apply our FastAttention to other NPUs and low-resource GPUs if accessible. We will also focus on compilation to facilitate easy invocation of our methods by users.

## 7 REPRODUCIBILITY STATEMENT.

In this work, we ensure the reproducibility of our results through detailed descriptions of the approaches and experimental setups. Besides the closed-source PanGu-series models, the experiments were conducted using the publicly available LLaMA-series models, and we specify the exact batch sizes, and sequence lengths used in our evaluations. The hardware configuration, including the 8 Ascend 910B NPUs and V100 GPUs, is clearly described to facilitate replication of the experiments. The scripts used to generate all results and performance metrics presented in the paper will be made available soon.

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

## A FLASHATTENTION SERIES

**FlashAttention:** As shown in Figure 12, FlashAttention employs the unified tiling strategy to split Query ($Q$), Key ($K$), and Value ($V$) into multiple blocks with small block size along the batch and heads dimension. Then, FlashAttention utilize online softmax (Milakov & Gimelshein, 2018; Rabe & Staats, 2021) and fused block GEMM to complete the attention calculation. To succinctly summarize, each of these $Q$ tiling blocks sequentially executed the following four stages: matrix multiplication of $Q * K^T$, $Exp$ calculation for softmax, multiplication of $Exp$ and $V$, and updates of the output. During the process, the martrix multiplication, i.e., GEMM, is handled by Tensor cores while non-GEMM operations like softmax are performed by CUDA Cores. This design minimizes memory access between SRAM and HBM while introduces frequent data flow between Tensor Cores and CUDA Cores due to the small block size imposed by SRAM limitations.

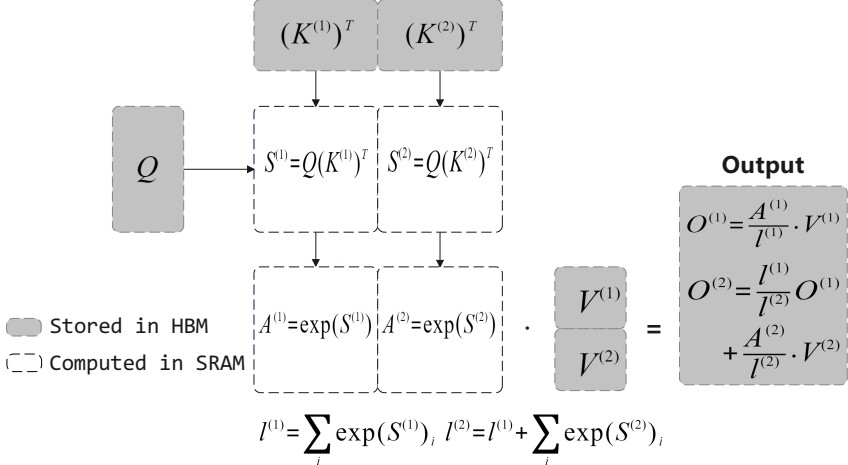

Figure 12: A streamlined depiction of the forward pass in FlashAttention.

**FlashAttention2:** FlashAttention2 refines the FlashAttention algorithm by reducing the number of non-matrix multiplication (non-matmul) FLOPs, while preserving the same output. It parallelizes both the forward and backward passes along the sequence length dimension, in addition to the batch and heads dimensions. This enhancement improves GPU resource utilization, particularly in cases where sequences are long and batch sizes are small. Additionally, within each block of attention computation, FlashAttention2 adjusts the cyclic order of operations to minimize communication and reduce SRAM reads and writes.

**FlashAttention3**: FlashAttention3 is specifically targeted on the newer GPU architectures, such as Hopper and Blackwell. FlashAttention3 introduces a restructured warp pipeline and enhances hardware utilization by overlapping the comparatively low-throughput non-GEMM operations, such as floating-point multiply-add and exponential computations, with asynchronous WGMMA instructions for GEMM execution. FlashAttention2 is regarded as the state-of-the-art (SOTA) method for the Ampere architecture, while FlashAttention3 represents the SOTA for architectures above Hopper.

## B THE DETAILED FASTATTENTION ALGORITHM FOR NPUS

Given the matrices $Q, K, V \in R^{B \times N \times S \times d}$ and $block\ sizes\ B_r$ and $B_c$, the $Q$ matrix will be splited along S dimension into $Q_1, Q_2, ..., Q_r \in R^{B_r \times d}$. In our two-level tiling strategy, the first level adapts the larger block size $B_r$. There are a total of $B \times N \times \lceil \frac{S}{B_r} \rceil$ large blocks and these blocks will be distributed across AI Cores. In each AI Core, it will follow the computations as blow:

$$Matrices \quad K, V \in R^{S \times d} \quad O_i, Q_i \in R^{B_r \times d} \quad (The\ First\ Level) \tag{1}$$

$$K, V : block\ size = B_c \quad K_1, ..., K_c\ and\ V_1, ..., V_c \in R^{B_c \times d} \tag{2}$$

$$Init : O_i^{(0)} \in R^{B_r \times d}, l_i^{(0)} \in R^{B_r}, m_i^{(0)} = (-\infty)_{B_r} \in R^{B_r} \tag{3}$$

$$for\ 1 \le j \le c : \tag{4}$$

$$S_i^{(j)} = Q_i K_j^T \in R^{B_r \times B_c}(Cube) \tag{5}$$

$$S_i^{(j)} : block\ size = B_b \quad S_{i1}, ..., S_{ib} \in R^{B_b \times B_c}(Second\ Level) \tag{6}$$

$$for\ 1 \le k \le b : \quad (Vector) \tag{7}$$

$$m_{ik}^{(j)} = max(m_{ik}^{(j-1)}, rowmax(S_{ik}^{(j)})) \in R^{B_b} \tag{8}$$

$$P_{ik}^{(j)} = exp(S_{ik}^{(j)} - m_{ik}^{(j)}) \in R^{B_b \times B_c} \tag{9}$$

$$l_{ik}^{(j)} = e^{m_{ik}^{(j-1)} - m_{ik}^{(j)}} l_{ik}^{(j-1)} + rowsum(P_{ik}^{(j)}) \in R^{B_b} \tag{10}$$

$$M_i^{(j)} = P_i^{(j)} V_j \in R^{B_r \times d} \quad (Cube) \tag{11}$$

$$O_i^{(j)} = diag(e^{m_i^{(j-1)} - m_i^{(j)}})^{-1} O_i^{(j-1)} + M_i^j \quad (Vector) \tag{12}$$

$$O_i = diag(l_i^c)^{-1} O_i^c \quad (Vector) \tag{13}$$

$$\tag{14}$$

In the tiling-AllReduce strategy, once the attention for $N$ heads in a sequence is completed, the large blocks proceed to perform the Linear and AllReduce operations:

$$O_i = attention(Q_i, K, V) \in R^{B_r \times Nd}, W_o \in R^{Nd \times H} \tag{15}$$

$$If\ this\ is\ the\ last\ block\ for\ the\ current\ sequence : \tag{16}$$

$$Linear\_out = OW_o \in R^{B_r \times H} \tag{17}$$

$$Final\_Out = Allreduce(Linear\_out) \tag{18}$$

$$\tag{19}$$

## C  THE DETAILED MECHANISM OF THE DATA LAYOUT REDESIGN

We here provide a detailed explanation of the data layout redesign for the Volta architecture. Fundamentally, a layout maps coordinate spaces to an index space. Layouts can be combined and manipulated to construct more complicated layouts and to tile layouts across other layouts. This can help users do things like partition layouts of data over layouts of threads. In the Cutlass library, a layout is a tuple of (Shape, Stride). Semantically, it implements a mapping from any coordinate within the Shape to an index via the Stride. For instance, Shape:(4,2) and Stride: (1,4) is a 4x2 column-major layout with stride-1 down the columns and stride-4 across the rows. And the more detailed introduction of layout can be found in the Cutlass documents.

Furthermore, high-end GPUs feature resource-rich architectures, such as Ampere and Hopper, which differ from Volta in terms of MMA instructions and thread-data layouts. The Cutlass documentation provides a comprehensive description of thread-data layouts for Volta, Ampere, and Hopper architectures. For the implementation of FastAttention, it is crucial to utilize the MMA instructions of the Volta architecture and redesign the data layout to support these instructions.

Specifically, there are two matrix multiplications in the workflow of attention. To clarify the challenges associated with adding support for Volta's m8n8k4 MMA instruction, we consider the *convert_layout_acc_Aregs* function, which converts the layout of the output argument C from the first matrix multiplication into the layout of the input argument A for the second multiplication.

When executing a single MMA instruction for matrices A, B, and C, the data is distributed across the threads. For Ampere's m16n8k16, as illustrated in Figure 13, thread 0 contains 8 elements of matrix $A$ (V0-V7), 4 elements of matrix $B$ (V0-V3), and produces 4 elements of matrix $C$ (V0-V3).

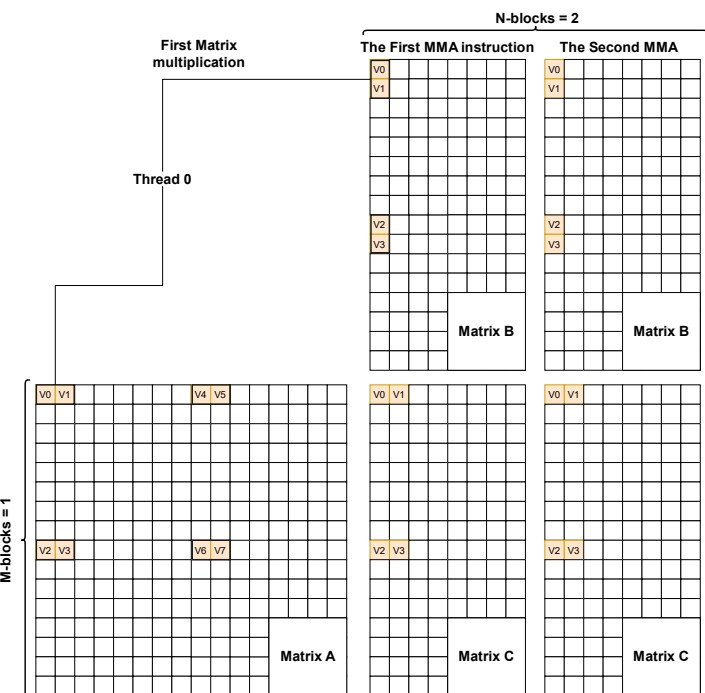

Figure 13: The first matrix multiplication using Ampere's m16n8k16 instruction.

These 4 elements of the matrix $C$ are located in the same places (row and column) what are the first 4 elements of the matrix $A$. This pattern holds true for the other threads as well. The first matrix multiplication requires two m16n8k16 MMA instructions to produce two matrices $C$. To utilize matrix C from the first multiplication as matrix $A$ for the second multiplication, the number of N must be even, allowing us to convert the CuTe layout of the two matrices $C$. This is the function of *convert_layout_acc_Aregs*.

For the Volta m8n8k4 instruction with FP32 accumulator, as shown in Figure 14, the thread0 contains 4 elements of matrix $A$(V0-V3), 4 elements of matrix $B$(V0-V3), and produces 8 elements of output matrix $C$(V0-V7). For the second matrix multiplication, two instructions must be executed, and the matrix $C$ of the first multiplication must be split into two martices as two matrices $A$ in the second multiplication. But half of the elements of the matrix $C$, that are in the registers of the thread0, are not the needed elements to perform the next matrix multiplication. And before performing the next multiplication, the threads need to exchange elements, which leads to synchronization and slowdown.

For faster back-to-back matrix multiplication, we use Volta m8n8k4 instruction with FP16 accumulator. To execute this instruction, each thread operates with the same number of elements as in the case of the FP32 accumulator, but uses another layout of elements that is shown in Figure 15. And in this case, the matrix $C$ of the first multiplication can be divided into two matrices $A$ of the second multiplication, without the need for the exchange between threads and synchronization.

To convert the layout of the argument $C$ of the first multiplication into the layout of argument $A$ of the second for any supported MMA, a converter has been developed that converts the layout in compile time depending on the CuTe *MMA_Traits* used.

## D  FORMULAS

Generative inference with LLMs typically consists of two stages: the *prefill* stage and the *decoding* stage. During the *prefill* stage, the key-value (KV) cache is generated with the prompt sequence. Afterwards, the *decoding* stage uses the generated KV cache to generate tokens one-by-one and meanwhile updates KV cache themselves.

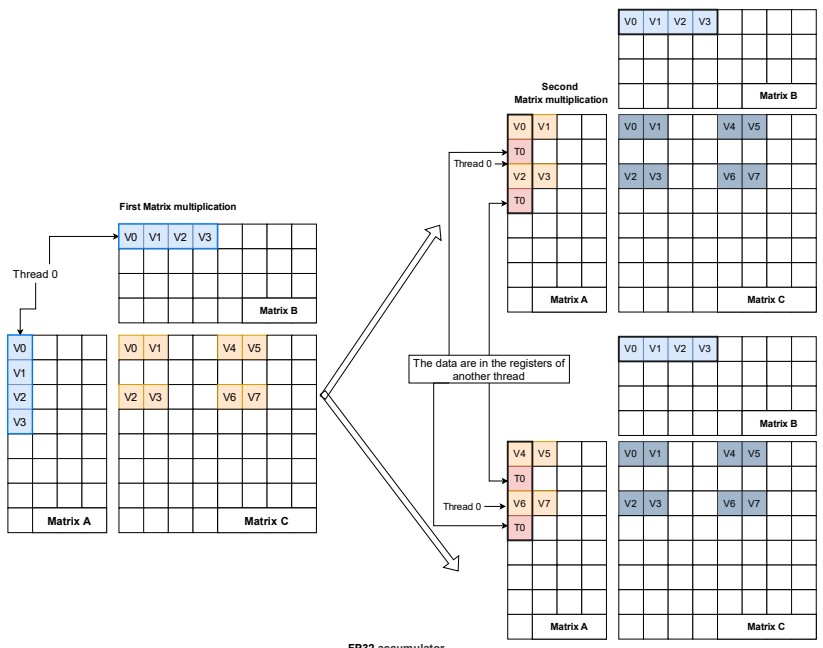

Figure 14: The execution of the two matrix multiplication using m8n8k4 instruction with FP32 accumulator.

Let $B$ represent the batch size. $S$ and $O$ represent the input and output sequence length, respectively. The hidden dimension of the attention layer is denoted as $H_1$, while the hidden dimension of the second MLP layer is $H_2$, and the total number of transformer layers is $L$. Denote the index of a transformer layer as $i$. The weight matrices of a transformer layer are denoted by $W_Q^i, W_K^i, W_V^i, W_O^i$, $W_1^i$ and $W_2^i$. Specifically, $W_Q^i, W_K^i, W_V^i, W_O^i \in \mathbb{R}^{H_1 \times H_1}$, $W_1^i \in \mathbb{R}^{H_1 \times H_2}$, and $W_2^i \in \mathbb{R}^{H_2 \times H_1}$.

For the *prefill* stage, $X^i$ represents the input matrix of the $i$-th transformer layer, and $X_Q^i, X_K^i$, $X_V^i, X_O^i$ is *query*, *key*, *value*, and *output* of the attention layer, respectively. All of them have dimensions $\mathbb{R}^{B \times S \times H_1}$. The specific computation of the $i$-th layer is as follows:

$$X_K^i = X^i \cdot W_K^i \tag{20}$$

$$X_V^i = X^i \cdot W_V^i \tag{21}$$

$$X_Q^i = X^i \cdot W_Q^i \tag{22}$$

$$X_O^i = f_{Softmax}\left(\frac{X_Q^i X_K^{i}{}^T}{\sqrt{h}}\right) \cdot X_V^i \cdot W_O^i + X^i \tag{23}$$

$$X^{i+1} = f_{act}(X_O^i \cdot W_1^i) \cdot W_2^i + X_O^i \tag{24}$$

For the *decoding* stage, denote the input matrix of the $i$-th layer by $T^i$. The *query*, *key*, *value*, and *output* of the $i$-th layer corresponding to the input $T^i$ are denoted as $T_Q^i, T_K^i, T_V^i, T_O^i$, respectively. Note that $T^i, T_Q^i, T_K^i, T_V^i, T_O^i \in \mathbf{R}^{B \times 1 \times H_1}$. The computation for the $i$-th layer is depicted as

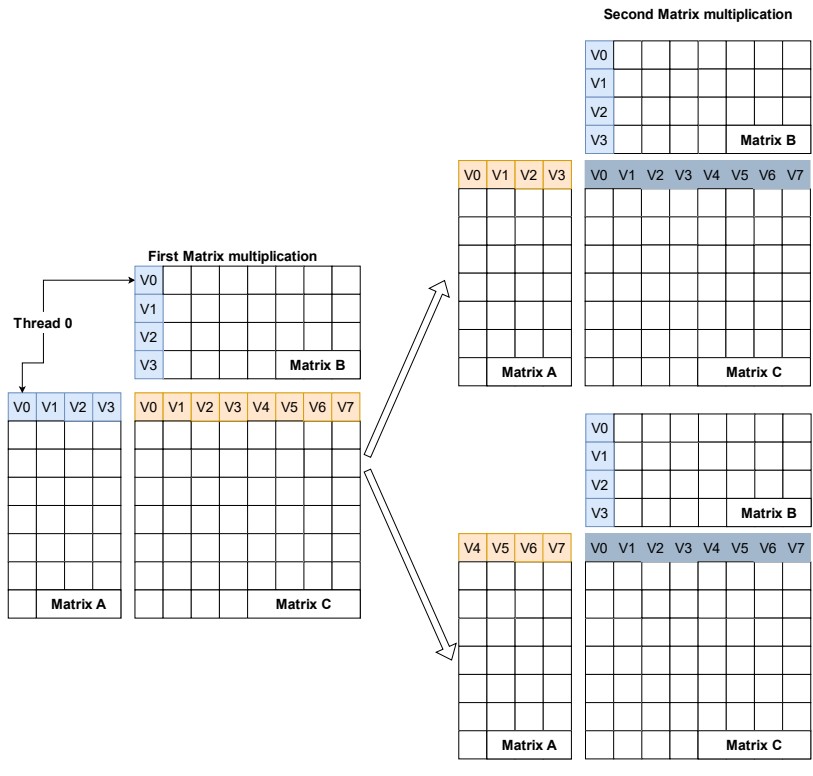

Figure 15: The execution of the two matrix multiplication using m8n8k4 instruction with FP16 accumulator.

follows:

$$T_K^i = T^i \cdot W_K^i \tag{25}$$

$$T_V^i = T^i \cdot W_V^i \tag{26}$$

$$X_K^i \leftarrow Contact(X_K^i, T_K^i) \tag{27}$$

$$X_V^i \leftarrow Contact(X_V^i, T_V^i) \tag{28}$$

$$T_Q^i = T^i \cdot W_Q^i \tag{29}$$

$$T_O^i = f_{Softmax}(\frac{T_Q^i {X_K^i}^T}{\sqrt{h}}) \cdot X_V^i \cdot W_O^i + T^i \tag{30}$$

$$T^{i+1} = f_{act}(T_O^i \cdot W_1^i) \cdot W_2^i + T_O^i \tag{31}$$

In our fine-grained CPU-GPU cooperative strategy, we can get the $L_{GPU}$ and $L_{CPU}$ by:

$$L_{GPU} = \frac{M_{GPU} - \frac{M_w}{n} - M_{mid} - M_{vocab}}{M_{kv}} \tag{32}$$

$$L_{CPU} = L - L_{GPU} \tag{33}$$

In detail, the GPU memory is primarily occupied by the model weights and the KV cache. The vocabulary matrix has the dimensions $\mathbf{R}^{V \times H_1}$, and $V$ is the size of the vocabulary. The max memory occupied by intermediate results can be from Equation 36. The memory footprint of the bias matrices is negligible. In case weights, KV cache, and others are stored in the `FP16` format, the Equation 32

can be extended as follows:

$$M_w = L(2 * 4 * H_1 * H_1 + 2 * 2 * H_1 * H_2)$$
$$= L(8H_1^2 + 4H_1H_2) \tag{34}$$

$$M_{kv} = \frac{2 * 2 * B * H_1(S + O)}{n}$$
$$= \frac{4BH_1(S + O)}{n} \tag{35}$$

$$M_{mid} = \frac{2 * 3 * B * S * H_1}{n}$$
$$= \frac{6BSH_1}{n} \tag{36}$$

$$L_{GPU} = \frac{M_{GPU} - \frac{M_w}{n} - M_{mid} - M_{vocab}}{M_{kv}}$$
$$= \frac{M_{GPU} - \frac{L(8H_1^2 + 4H_1H_2)}{n} - \frac{6BSH_1}{n} - VH_1}{\frac{4BH_1(S + O)}{n}} \tag{37}$$
$$= \frac{nM_{GPU} - L(8H_1^2 + 4H_1H_2) - 6BSH_1 - nVH_1}{4BH_1(S + O)}$$

# E    FULL EXPERIMENTS RESULTS

## E.1    PERFORMANCE EVALUATION ON VISION TRANSFORMERS

FastAttention targets the scenarios where the attention module constitutes a significant portion of the overall computational time during model inference. In particular, attention is not a bottleneck for Vision and Diffusion Transformers, as shown in Table 7. For instance, the attention only takes 4% of total times for ViT-B inference. That's why we disregard Vision or Diffusion transformer as baseline. Despite the fact, we still test the single-operator speedups of FastAttention over the standard attention to illustrate the effectiveness of FastAttention for the attention calculation. We evaluate FastAttention using DeiT-B model with varying batchsizes. The results are shown in Table 8. Our FastAttention achieves a speedup ranging from $2.52\times$ to $7.58\times$ as the batch size increases from 32 to 1024. However, this improvement has a negligible impact on the end-to-end network speedups.

Table 7: Time complexity and computation breakdown of ViT and DeiT.

| Model | QKV projection | Attention | O project | MLP | others |
|-------|----------------|-----------|-----------|-----|--------|
| ViT-B/384 | 22% | 11% | 7% | 59% | 1% |
| ViT-B | 24% | 4% | 8% | 63% | 1% |
| DeiT-S | 23% | 8% | 8% | 61% | 1% |
| DeiT-Ti | 21% | 14% | 7% | 56% | 2% |

## E.2    QUANTIZATION

Our FastAttention is orthogonal to general hardware-agnostic approaches such as quantization, pruning, and distillation. For instance, Table 9 compares FastAttention with FP16 and naive INT8 precisions using PanGu-71B, demonstrating that FastAttention can be used alongside other compression methods to further enhance inference performance. With a batch size of 1 and varying sequence lengths, FastAttention achieves a speedup of approximately $1.2\times$ when used alongside quantization techniques.

## E.3    TILING-ALLREDUCE EVALUATION

We conducted additional experiments to further demonstrate the efficiency of the proposed tiling-AllReduce strategy. Specifically, we compared the latency of FastAttention with and without the

Table 8: Single-operator Performance of FastAttention using Deit-B models' dimensions on an Ascend 910B.

| Batchsize | Standard attention(ms) | FastAttention(ms) | Speenup |
|---|---|---|---|
| 32 | 1.21 | 0.48 | 2.52× |
| 64 | 3.05 | 0.66 | 4.62× |
| 128 | 6.14 | 1.08 | 5.68× |
| 256 | 12.183 | 1.828 | 6.664× |
| 512 | 24.25 | 3.52 | 6.89× |
| 1024 | 48.40 | 6.38 | 7.58× |

Table 9: The performance evaluation of FastAttention using FP16 and INT8

| Model | seq_length | Latency(us)-FP16 | Latency(us)-INT8 | Speedup |
|---|---|---|---|---|
| PanGu-71B | 128 | 55.01 | 42.77 | 1.286× |
| PanGu-71B | 256 | 58.84 | 50.99 | 1.153× |
| PanGu-71B | 512 | 56.65 | 57.4 | 0.987× |
| PanGu-71B | 1K | 77.77 | 62.36 | 1.247× |
| PanGu-71B | 2K | 113.49 | 93.43 | 1.214× |
| PanGu-71B | 4K | 279.94 | 222.325 | 1.26× |

tiling-AllReduce strategy. The configuration without the tiling-AllReduce strategy involves the unfused FastAttention kernel (as described in § 4.1 ), *Linear* operator, and the `Allreduce` operation. The evaluations were carried out using the PanGu-38B model with varying batch sizes and sequence lengths across 8 Ascend 910B NPUs. we similarly observed that the tiling-AllReduce strategy enabled FastAttention to achieve speedups ranging from 1.2× to 1.5×. Additionally, we measured runtime performance by varying the sequence length from 1K to 32K while adjusting the batch size to ensure a total token count of 32K. The results, reported in Figure 16, show that the tiling-AllReduce strategy allows FastAttention to achieve up to a 1.53× speedup, demonstrating significant performance improvements.The experiments demonstrate that our FastAttention achieves significant performance improvements regardless of changes in batch size or sequence length.

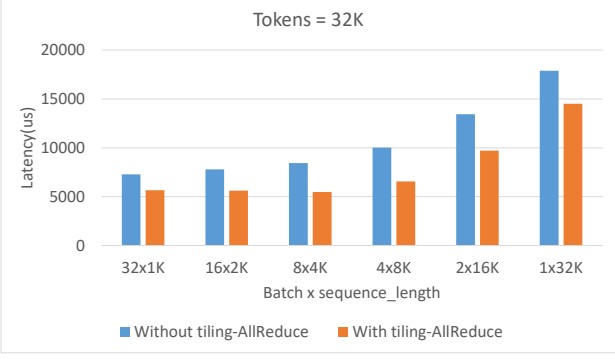

Figure 16: The performance evaluation of tiling-AllReduce strategy with 32K tokens.

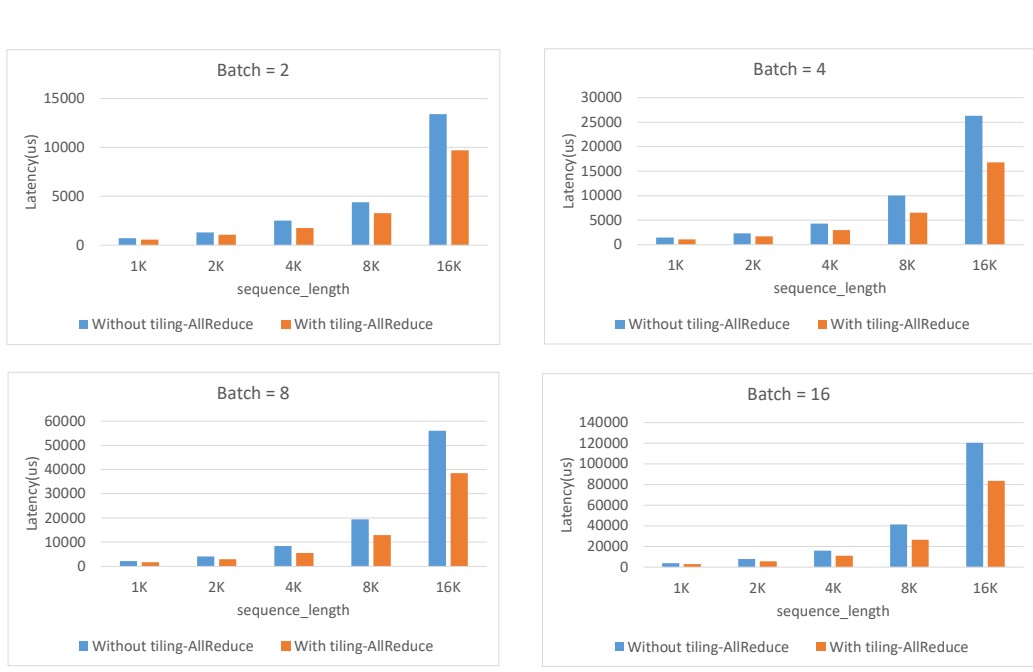

Figure 17: The performance evaluation of FastAttention with/without tiling-AllReduce strategy on 8 Ascend 910B.

