# OpenReview forum: "FastAttention: Extend FlashAttention2 to NPUs and Low-resource GPUs for Efficient Inference"
_ICLR.cc/2025/Conference — Submitted to ICLR 2025_

### Official Review · Reviewer_8h42 · 2024-11-02

**Soundness:** 3
**Presentation:** 2
**Contribution:** 3
**Rating:** 6
**Confidence:** 3

**Summary:**

Existing implementations of FlashAttention do not support older or low-resource GPUs, such as those with Volta architecture and earlier models, as well as NPUs. This paper introduces FastAttention, the first adaptation of FlashAttention designed for these types of accelerators. It outlines the challenges and opportunities associated with porting FlashAttention to NPUs and pre-Ampere NVIDIA GPUs. The authors propose an implementation that optimizes performance by leveraging the specific hardware features of the target NPU or pre-Ampere GPU. Additionally, the evaluation demonstrates the performance of FastAttention in both single and multi-accelerator scenarios.

**Strengths:**

- The ability to run the faster FlashAttention on a greater number of AI accelerators is an important contribution to the AI community.
- A well-designed solution has been proposed to effectively utilize the Cube/Vector units and the memory hierarchy in NPUs.
- There is a clear comparison of the use of the Matrix Multiply Accumulator (MMA) in the Volta architecture versus later architectures.
- The evaluation is thorough and comprehensive.

**Weaknesses:**

- The work appears to focus more on incremental implementation rather than addressing or solving a novel problem innovatively. To enhance this work, the author could investigate how the techniques used to adapt the flashattention kernels for NPUs and low-resource GPUs might be applicable to a broader range of kernels designed for Turing, Ampere, and Blackwell architectures. For instance, any kernels based on Matrix Multiply Accumulation (MMA) could also benefit from these adaptations. In this context, flashattention could serve as a significant practical example that is explored in detail within the paper.
- The contributions seem to be limited to a few specific architectures. For example, although the title mentions "low-resource GPUs," FastAttention appears to primarily support V100 GPUs. Does FastAttention support all NVIDIA Volta GPUs? What about Pascal or earlier architectures? Additionally, what is the support status for non-NVIDIA architectures? It would be helpful to clearly specify which hardware architectures are supported, even if some are not.
- The paper discusses only the inference scenario, but it is unclear whether FastAttention is designed solely for inference or if it also supports training. The authors should clarify this and, if training is supported, include a discussion in the paper. For example, what are the memory requirements during backpropagation and how do the proposed optimizations affect gradient computation?
- The prior work and background on the fusion of attention and linear calculations in Section 4.2 could be explained more clearly and supplemented with relevant citations.

**Questions:**

- Does FastAttention work with any type/version of NPUs, or is it limited to only certain models?
- Is FastAttention compatible with low-resource GPUs beyond the ones mentioned in V100 and those based on the Volta architecture?
- Could you provide more details about the implementation of FastAttention? Was the code built on existing libraries or repositories? Is there a plan to integrate it into existing open-source LLM frameworks?
- In section 4.1, you describe an optimization intended to reduce the memory requirement for the causal mask. However, since the causal masking is determined solely by the relative positions in the sequence for each attention score, why generate a causal matrix? Could this be achieved with a simple instruction in the kernel that sets the result to zero for items that should not be included? To enhance the paper, it would be beneficial to assess this option in the experiments.
- Could you enhance the figure captions so that all relevant details of the experiments can be understood simply by looking at the plots and their captions? At times, readers need to refer back to the main text to grasp the specifics of an experiment, such as whether it measures prefill versus decode latency or the parallelization strategy employed. Including key takeaways in the figure captions would also be beneficial.

---

> ### Author Response · Authors · 2024-11-22
> **Response to Reviewer 8h42 (1/3)**
>
> Dear Reviewer 8h42,
>
> Thank you for your detailed review and thoughtful feedback on our manuscript. We greatly appreciate your recognition of the strengths in our work, especially the innovative aspects of our methodology and the thoroughness of our experimental evaluation. We are eager to address your concerns and questions, as outlined below:
>
> **Weakness 1:** The work appears to focus more on incremental implementation rather than addressing or solving a novel problem innovatively. To enhance this work, the author could investigate how the techniques used to adapt the flashattention kernels for NPUs and low-resource GPUs might be applicable to a broader range of kernels designed for Turing, Ampere, and Blackwell architectures. For instance, any kernels based on Matrix Multiply Accumulation (MMA) could also benefit from these adaptations. In this context, flashattention could serve as a significant practical example that is explored in detail within the paper.
>
> **Reply:** Thanks. We would like to highlight the significant differences between our FastAttention and similar works. FlashAttention employs the tiling and online softmax methods to reduce memory and I/O complexity for modern CUDA architecture, such as Ampere and Hopper. FastAttention for NPUs, while also adopting tiling and online softmax strategies, introduces several significant differences that distinguish it from both FlashAttention and memory-efficient Attention:
> - **Two-level tiling:** FlashAttention leverages the high-bandwidth on-chip memory, i.e., shared memory in CUDA architectures, to significantly reduce I/O overhead. In contrast, the L1 cache in Ascend NPUs is decoupled, making FlashAttention's method less efficient in terms of hardware utilization. Consequently, we propose two-level tiling strategy to solve the challenge.
>     1. **Overlapping:** FastAttention is optimized for decoupled architectures and achieves greater efficiency by overlapping the GEMM (General Matrix Multiplication) operations, performed by the Cube unit, with element-wise calculations (e.g., softmax), handled by the Vector unit.  In contrast, FlashAttention series are not conveniently supportive to this kind of pipelines.
>     2. **Cache level:** FastAttention encourages the assignment of larger block sizes to the Cube unit to fully exploit its computational power, while smaller block sizes should be allocated to the Vector unit to better fit varying L1 buffer sizes and reduce synchronization overhead between the Cube and Vector units. In contrast, FlashAttention employs a smaller block size for both Tensor Cores and CUDA Cores.
>     3. **Row-wise Partitioning:** Additionally, the Vector unit splits large block matrices along the row dimension to minimize the number of updates (e.g., rowmax, $l$, and the $P$ matrix) required during softmax computation, whereas FlashAttention does not.
> - **Tiling-AllReduce strategy:** FastAttention employs a tiling-AllReduce strategy to overlap computations with AllReduce communication during inference with Tensor Parallelism (TP), further improving efficiency. However, FlashAttention series lacks the feature.
> - **Tiling-mask:** We propose an architecture-agnostic tiling-mask strategy to eliminate the memory requirement for $attention\\_mask$ matricx. Note that the $attention\\_mask$ matrix is indispensable for the architectures employing the SIMD model, such as Ascend NPUs.
>
> The high-end GPUs, such as the A100 and H100, are not only costly but also face significant supply shortages, making them increasingly difficult to acquire. As a result, many academic institutions and research organizations, particularly in developing regions, are unable to access the high-end GPUs necessary for cutting-edge research. Given these constraints, adapting state-of-the-art attention mechanisms to run efficiently on low-resource hardware is an essential challenge for advancing research in the field.
>
> Besides, we highly appreciate your suggestion and we will explore how the techniques developed for adapting FlashAttention kernels to NPUs and low-resource GPUs could be generalized to a wider range of kernels designed for modern CUDA architectures in future work.

---

> ### Author Response · Authors · 2024-11-22
> **Response to Reviewer 8h42 (2/3)**
>
> **Weakness 2:** The contributions seem to be limited to a few specific architectures. For example, although the title mentions "low-resource GPUs," FastAttention appears to primarily support V100 GPUs. Does FastAttention support all NVIDIA Volta GPUs? What about Pascal or earlier architectures? Additionally, what is the support status for non-NVIDIA architectures?  Is FastAttention compatible with low-resource GPUs beyond the ones mentioned in V100 and those based on the Volta architecture?
>
> **Reply:** Yes, FastAttention supports the NVIDIA GPUs with Volta architecture. Actually, our design is specifically targeted at Volta GPUs. This is because Volta architecture GPU is the only type of GPU that needs to be supported. On one hand, higher-level architectures like Ampere and Hopper already have efficient attention implementations, i.e., FlashAttention series. On the other hand, GPUs with lower-level architectures than Volta, such as Pascal and Kepler, do not possess Tensor Cores, which is a must for implementing various efficient attention. Moreover, given the scarcity of obtaining high-end GPUs like H100 nowadays, Volta GPUs are still widely utilized in many regions of the world for its ultra-high cost-effectiveness.
>
> For non-NVIDIA architectures, FastAttention is compatible with all Ascend NPUs featuring a decoupled architecture. The strategies proposed in FastAttention are general and applicable to similar architectures.
>
> **Weakness 3:** The paper discusses only the inference scenario, but it is unclear whether FastAttention is designed solely for inference or if it also supports training. The authors should clarify this. For example, what are the memory requirements during backpropagation and how do the proposed optimizations affect gradient computation?
>
> **Reply:** As indicated by the title of our paper, we focus solely on efficient inference in this study. We believe that our contributions are adequate even if we only focus on inference. Actually, both training and inference are typical scenarios for efficient attention. Moreover, compared with training that is usually performed once, inference is more frequent in daily life for an LLM. Optimization for inference scenarios is profoundly beneficial in promoting AI democracy. However, your suggestion is highly valued and we put the training acceleration using FastAttention as our future work.
>
> **Weakness 4:** The prior work and background on the fusion of attention and linear calculations in Section 4.2 could be explained more clearly and supplemented with relevant citations.
>
> **Reply:** For the self-attention calculation: $x_{out} = Softmax(\frac{QK^T}{\sqrt{d}})·V·W_o + x$,
> FlashAttention series accelerates the calculation of $x_o=Softmax(\frac{QK^T}{\sqrt{d}})·V$, the Linear calculation implys the multiplication of $x_o$ and the $W_o$ matrix.
>
> For the fusion of attention and linear calculation, we here provide a detailed description mathematically with equations. Given the matrices $Q,K,V \in R^{B \times N \times S \times d}$ and $block \; sizes \; B_r$ and $B_c$, the $Q$ matrix will be splited along S dimension into $Q_1, Q_2, ... , Q_r \in R^{B_r \times d}$. In our two-level tiling strategy, the first level adapts the larger block size $B_r$. There are a total of $B \times N \times \lceil \frac{S}{B_r} \rceil$ large blocks and these blocks will be distributed across AI Cores.
> In each AI Core, it will follow the computations as blow:
>
> $Matrices \quad K,V \in R^{S \times d} \quad O_i,Q_i \in R^{B_r \times d} \quad (The \ First \ Level)$ \
> $K,V: block \, size = B_c \quad K_{1},...,K_{c} \, and \, V_{1},...,V_{c}\in R^{B_c \times d}$ \
> $Init: O_{i}^{(0)} \in R^{B_r \times d},l_i^{(0)} \in R^{B_r}, m_i^{(0)} =(-\infty)_{B_r} \in R^{B_r}$
>
> $for \ 1 \leq j \leq c:$ \
> $\qquad  S_i^{(j)} = Q_{i}K_{j}^T \in R^{B_r \times B_c} (Cube)$ \
> $\qquad  S_i^{(j)}: block \ size=B_b \quad S_{i1},...,S_{ib} \in R^{B_b \times B_c} (Second \ Level)$ \
> $\qquad  for \: 1 \leq k \leq b: \qquad (Vector)$ \
> $\qquad \qquad  m_{ik}^{(j)} = max(m_{ik}^{(j-1)},rowmax(S_{ik}^{(j)})) \in R^{B_b}$ \
> $\qquad \qquad  P_{ik}^{(j)} = exp(S_{ik}^{(j)} - m_{ik}^{(j)}) \in R^{B_b \times B_c}$ \
> $\qquad \qquad  l_{ik}^{(j)} = e^{m_{ik}^{(j-1)}-m_{ik}^{(j)}} \, l_{ik}^{(j-1)} +rowsum(P_{ik}^{(j)}) \in R^{B_b}$ \
> $\qquad M_{i}^{(j)} = P_{i}^{(j)}V_{j} \in R^{B_r \times d}\quad(Cube)$ \
> $\qquad O_{i}^{(j)} = diag(e^{m_{i}^{(j-1)}-m_{i}^{(j)}})^{-1}O_{i}^{(j-1)} + M_{i}^j \quad(Vector)$ \
> $O_{i} = diag(l_i^c)^{-1}O_{i}^c \quad (Vector)$
>
> In the tiling-AllReduce strategy, once the attention for $𝑁$ heads in a sequence is completed, the large blocks proceed to perform the Linear and AllReduce operations:
>
> $O_i = attention(Q_i,K,V)\in R^{B_r \times Nd}, W_o \in R^{Nd \times H}$ \
> $If \ this \ is \ the \ last \ block \ for \ the \ current \ sequence:$ \
> $\qquad Linear\\_out = OW_o \in R^{B_r \times H}$ \
> $\qquad Final\\_Out = Allreduce(Linear\\_out)$

---

> ### Author Response · Authors · 2024-11-22
> **Response to Reviewer 8h42 (3/3)**
>
> **Question 1:** Does FastAttention work with any type/version of NPUs, or is it limited to only certain models?
>
> **Reply:** FastAttention is compatible with all Ascend NPUs featuring a decoupled architecture. Similar to FlashAttention, FastAttention is applicable to any large models that utilize attention mechanisms. In short, our FastAttention possesses the identical model application range as FlashAttention. Morevoer, our FastAttention offers a simple usage interface. For example:
> ```
> //FlashAttention2
> from flash_attn import flash_attn_func
> output = flash_attn_func(Q,K,V)
> //VERSUS FastAttention
> from fastattention import fast_attn_func
> output = fast_attn_func(Q,K,V)
> ```
>
> **Question 2:** Could you provide more details about the implementation of FastAttention? Was the code built on existing libraries or repositories? Is there a plan to integrate it into existing open-source LLM frameworks?
>
> **Reply:** We plan to integrate our work into PyTorch and the implementation details of FastAttention are summarized in the table below.
>
> ||FastAttention on NPUs|FastAttention on Volta GPUs|
> |:-:|:-:|:-:|
> |Supported hardware|Ascend NPUs featuring decopuled architecture,such as Ascend 910B|Volta GPUs such as Tesla V100 and Titan V|
> |Dependencies|CANN(Compute Architecture for Neural Networks), HCCL(Huawei Collective Communication Library), AOL(Ascend Operator Library), Ascend C|CuTe, Cutlass(Compute Architecture for Neural Networks),CUDA(Compute Unified Device Architecture)|
> |Supported framework|Pytorch|Pytorch|
> |Precision Support|FP32,FP16,INT8|FP16|
> |Memory Complexity|$O(N)$|$O(N)$|
> |Supported max sequence length with PanGu-38B|128K on 8 Ascend 910B NPUs|256K on 8 V100 GPUs|
> |Performance(max)|238.2TFLOPS|49.6TFLOPS|
>
> **Question 3:** In section 4.1, you describe an optimization intended to reduce the memory requirement for the causal mask. However, since the causal masking is determined solely by the relative positions in the sequence for each attention score, why generate a causal matrix? Could this be achieved with a simple instruction in the kernel that sets the result to zero for items that should not be included? To enhance the paper, it would be beneficial to assess this option in the experiments.
>
> **Reply:** The $attention\\_mask$ matrix is indispensable for NPUs. This is because CUDA architectures operate with the SIMT (Single Instruction, Multiple Threads) model, whereas NPUs adopt the SIMD (Single Instruction, Multiple Data) model.
>
> In the SIMT model, causal masking is determined solely by the relative positions in the sequence for each attention score. However, implementing this approach in the SIMD model, such as through a `for` loop, can be highly inefficient.
>
> **Question 4:** Could you enhance the figure captions so that all relevant details of the experiments can be understood simply by looking at the plots and their captions? At times, readers need to refer back to the main text to grasp the specifics of an experiment, such as whether it measures prefill versus decode latency or the parallelization strategy employed. Including key takeaways in the figure captions would also be beneficial.
>
> **Reply:** Thank you for your advice and kind reminder. We have enhanced the figure captions in our revised version.

---

> ### Author Response · Authors · 2024-11-25
> **Hope for the feedback**
>
> Dear Reviewer 8h42,
>
> Thanks for your valuable time and insightful comments. We deeply appreciate your constructive suggestions and have worked to incorporate them into the revised version. We hope the updates effectively address the concerns raised in your initial reviews and look forward to any further suggestions you may have for refining the manuscript.
>
> As the deadline for the Author/Reviewer discussion is approaching, please let us know if you require additional details or further clarifications from our side. We are fully committed to refining our work and are eager to engage in further discussions to enhance the quality of the submission. Once again, thank you for your kind consideration and guidance!

---

> ### Author Response · Authors · 2024-11-27
> **Looking forward to your reply**
>
> Dear Reviewer 8h42,
>
> Thank you again for your comments. Your opinion is highly valued, and we have been committed to providing comprehensive responses. We sincerely hope our efforts to address your concerns. We are delighted to provide any additional data, explanations, or results to further address your concerns at any time. We look forward to your feedback and hope for a positive outcome. Thank you very much for your time and consideration.

---

> ### Author Response · Authors · 2024-11-28
> **Reminder for Feedback**
>
> Dear Reviewer 8h42,
>
> As the deadline for submitting the revised PDF is only a few hours away, it may not be feasible to incorporate further changes into the current version. We apologize for this constraint at the final stage. However, we are fully committed to addressing any additional questions or concerns leading up to December 3rd.
>
> Here are some important deadlines to keep in mind:
>
> November 27th, 11:59 PM AoE: Last day for authors to upload a revised PDF. After this deadline, no further updates to the manuscript will be possible, and authors will only be able to respond to comments on the forum. If you’d like any changes reflected in the revised manuscript, please inform us before this time.
>
> December 2nd: Last day for reviewers to post messages to the authors (six-day extension). This is the final opportunity to share any remaining concerns with us.
>
> December 3rd: Last day for authors to post messages on the forum (six-day extension). After this date, we will no longer be able to respond to any concerns or feedback.
>
> We sincerely thank you once again for your time, effort, and valuable feedback, which have been instrumental in improving our work!

---

> ### Author Response · Authors · 2024-11-30
> **Any further suggestions or questions?**
>
> Dear Reviewer 8h42,
>
> We would like to express our sincere gratitude for the time and effort you have devoted to reviewing our work. We understand how demanding this time can be, especially with your own commitments, and truly appreciate the thoughtful attention you have given to our paper.
>
> We are deeply excited about this paper and its findings, and we greatly value the opportunity to engage in meaningful discussions with you. Please feel free to reach out with any questions, and we are happy to provide further clarifications.

---

> ### Author Response · Authors · 2024-12-02
> **Kind Reminder: Last Day of Reviewer-Author Discussion**
>
> Dear Reviewer 8h42,
>
> Thank you once again for your efforts and thoughtful comments. With only 24 hours remaining in the Author/Reviewer discussion period, we kindly ask if you could review our responses to your concerns and let us know if there are any additional questions or unresolved points. We would be happy to address them promptly.
>
> If you find our responses satisfactory, we would greatly appreciate it if you could consider reflecting this in your final score. Your valuable feedback is instrumental in improving the quality of our work, and we sincerely thank you for your contributions to this process.
>
> Best regards,
>
> The authors of Submission 9395

---

### Official Review · Reviewer_XLQ8 · 2024-11-03

**Soundness:** 2
**Presentation:** 3
**Contribution:** 2
**Rating:** 6
**Confidence:** 4

**Summary:**

This paper presents the FlashAttention (or memory-efficient Attention) on the NPU and implements it on the currently unsupported GPUs. It adapts the FlashAttention algorithm to NPU with a two-level tiling, presents the software pipeline of computation/communication on multi-NPUs, implements it on V100 GPU, and presents the solution of long context.

**Strengths:**

- Design/implement a critical Algorithm onto the new architecture. The pipeline and offloading method are not explored in the official FlashAttention.
- Achieve good speedup.

**Weaknesses:**

- Even though it presents solid implementation, the research contributed could be highlighted more. For example, the fusion and tiling of the Attention does not show significant difference to FlashAttention and memory-efficient Attention.
- It lacks the credit to memory-efficient Attention (Self-attention Does Not Need O(n^2) Memory), which is a concurrent (or earlier) work of fusing Attention with the similar method with FlashAttention-2, and supports TPU.

**Questions:**

- It claims in the introduction that, the existing FlashAttention cannot run on non-CUDA architectures. However, the memory-efficient Attention has supported TPU ((Self-attention Does Not Need O(n^2) Memory)), and its paper is released on 2021 Dec. Besides, there is also AMD GPU suppported FlashAttention (https://rocm.blogs.amd.com/artificial-intelligence/flash-attention/README.html), which is also non-CUDA.
- Section 4.2 describes the two-level tiling, is it the same with the normal GEMM implement on the NPU? This is similar to the two-level tiling on the GPU: block level and warp/MMA level.  A comparison between the two-level tiling described in this paper and the GEMM implementation can be helpful to show the unique contribution of this design.
- The current description does not highlight the research challenge of supporting FlashAttention on the V100 besides the engineer problems. Highlighting the unique design difference can better make the reader understand the contribution. Besides, supporting newer architecture rather than older architecture is the common research trend. Some discussion of supporting the old architecture could be helpful for this paper.

---

> ### Author Response · Authors · 2024-11-22
> **Response to Reviewer XLQ8 (1/2)**
>
> Dear Reviewer XLQ8,
>
> We sincerely appreciate your thoughtful evaluation and constructive feedback on our work. Your insights have been instrumental in guiding our revisions and enhancements. We address your concerns and questions as follows:
>
> **Weakness 1:** Even though it presents solid implementation, the research contributed could be highlighted more. For example, the fusion and tiling of the Attention does not show significant difference to FlashAttention and memory-efficient Attention.
>
> **Reply:**
> - **Memory-efficient attention** is a general algorithm that leverages a tiling method to reduce memory complexity from $O(N^2)$ to $O(N)$, ensuring compatibility with diverse hardware platforms, including TPUs. However, it may sacrifice some hardware-specific optimizations.
> - **FlashAttention** employs a similar algorithm to memory-efficient Attention and also uses tiling to reduce memory complexity. In contrast, FlashAttention is highly optimized for modern CUDA architectures, such as Ampere and Hopper, and significantly **reduces I/O complexity**, leading to more efficient computations compared to memory-efficient Attention.
> - **FastAttention**, while also adopting tiling and online softmax strategies, introduces several significant differences that distinguish it from both FlashAttention and memory-efficient Attention:
>     - **Overlapping:** FastAttention is optimized for decoupled architectures and achieves greater efficiency by overlapping the GEMM (General Matrix Multiplication) operations, performed by the Cube unit, with element-wise calculations (e.g., softmax), handled by the Vector unit.
>     - **Two-level tiling strategy:** FastAttention encourages the assignment of larger block sizes to the Cube unit to fully exploit its computational power, while smaller block sizes should be allocated to the Vector unit to better fit varying L1 buffer sizes and reduce synchronization overhead between the Cube and Vector units. In contrast, FlashAttention employs a smaller block size for both Tensor Cores and CUDA Cores.
>     - **Row-wise Partitioning:** Additionally, the Vector unit splits large block matrices along the row dimension to minimize the number of updates (e.g., rowmax, $l$, and the $P$ matrix) required during softmax computation.
>     - **Tiling-AllReduce strategy:** FastAttention employs a tiling-AllReduce strategy to overlap computations with AllReduce communication during inference with Tensor Parallelism (TP), further improving efficiency.
>
> **Weakness 2:** It lacks the credit to memory-efficient Attention (Self-attention Does Not Need O(n^2) Memory), which is a concurrent (or earlier) work of fusing Attention with the similar method with FlashAttention-2, and supports TPU.
>
> **Reply:** Thank you for your feedback. Memory-efficient Attention (Self-attention Does Not Need $O(N^2)$ Memory) is implemented in xFormers, which supports V100 GPUs. Consequently, we included citations to memory-efficient Attention in the references to xFormers, which led to some ambiguity in the citations. In the revised version, we have provided appropriate citations to this work. The modifications in this section have been highlighted with blue color.
>
> **Question 1:** It claims in the introduction that, the existing FlashAttention cannot run on non-CUDA architectures. However, the memory-efficient Attention has supported TPU ((Self-attention Does Not Need O(n^2) Memory)), and its paper is released on 2021 Dec. Besides, there is also AMD GPU suppported FlashAttention (https://rocm.blogs.amd.com/artificial-intelligence/flash-attention/README.html), which is also non-CUDA.
>
> **Reply:** Thank you very much for your kind reminder. We apologize for our vague claim. Our this claim essentially is supposed to deliver a opinion that FlashAttention did not adapt to NPUs and to achieve the adaption is non-trivial. We have optimized our claim and involved the work of AMD GPU FlashAttention in our paper.

---

> > ### Comment · Reviewer_XLQ8 · 2024-11-26
> >
> > Thanks for the clarification. I can figure out it requires a lot of effort to implement FlashAttention on the NPU. It is a great contribution from the aspect of the production. However, the unique research contribution is still not clear to me. It could be better to abstract the detailed adaption into some high-level and general insights that can be applied to a broader range (for example, the insight of memory-efficient attention can be applied to various devices).

---

> ### Author Response · Authors · 2024-11-22
> **Response to Reviewer XLQ8 (2/2)**
>
> **Questions 2:** Section 4.2 describes the two-level tiling, is it the same with the normal GEMM implement on the NPU? This is similar to the two-level tiling on the GPU: block level and warp/MMA level. A comparison between the two-level tiling described in this paper and the GEMM implementation can be helpful to show the unique contribution of this design.
>
> **Reply:** Our two-level tiling strategy is fundamentally different from the standard GEMM implementation on NPUs. The two-level tiling strategy addresses the synchronization overhead between the Cube and Vector units, specifically targeting the coordination between GEMM and element-wise operations. This strategy enables GEMM to overlap with element-wise operations, effectively reducing computation latency. In contrast, the normal GEMM implementation on NPUs is purely focused on matrix operations, dividing the matrix into multiple blocks based on the number of AI Cores and the size of the L1 buffer.
>
> We mark three levels of siginificant differences in tiling strategies between two-level tiling strategy and the tiling method of the normal GEMM on the NPU:
> 1. **Pipeline level.** Due to the benefits of decoupled architectures of NPUs, our strategy features a fine-grained pipeline between the Cube unit and the Vector unit, which **naturally allows the efficient overlap of the softmax computation with the GEMM (General Matrix Multiplication)**. In contrast, the standard GEMM implementation on NPUs is solely focused on matrix operations, which are handled exclusively by the Cube unit.
> 2. **Cache level.** Our two-level tiling strategy encourages the assignment of a larger block size to the Cube unit to fully leverage its computational power, while a smaller block size should be assigned to the Vector unit to accommodate the varying L1 buffer sizes and reduce synchronization overhead between the Cube and Vector units. In contrast, the standard implementation utilizes only the L1 buffer in the Cube unit and does not account for the synchronization overhead between the Cube and Vector units.
> 3. **Computation level.** Moreover, our FastAttention can split the matrix along the row dimension for the Vector unit to reduce the number of updates (e.g., rowmax and the P matrix) during the softmax computation, whereas the standard GEMM implementation does not consider this.
>
> **Question 3:** The current description does not highlight the research challenge of supporting FlashAttention on the V100 besides the engineer problems. Highlighting the unique design difference can better make the reader understand the contribution.
>
> **Reply:** The key research challenges in supporting FlashAttention on the V100 lies in the non-trivial differences in instruction sets and data layouts. Due to these discrepancies, we designed to execute two consecutive Volta's MMA operations using only registers without storing intermediate results. By redesigning the data layout, we successfully implemented this approach. Additional details can be found in the Appendix.
>
> **Question 4:** Besides, supporting newer architecture rather than older architecture is the common research trend. Some discussion of supporting the old architecture could be helpful for this paper.
>
> **Reply:** First of all, FastAttention can greatly enhance the application and speed of LLM inference on low-end GPUs and NPUs, potentially extending the use of LLMs to edge devices. Besides, high-end GPUs such as the A100 and H100 face severe supply shortages and are prohibitively expensive, making them increasingly inaccessible [1,2,3,4,5,6]. As highlighted in [5], "There is no sign that the GPU shortage we have today will abate in the near future."
>
> As a result, many academic institutions and research organizations, particularly in developing regions, are unable to access the high-end GPUs necessary for cutting-edge research. This widespread scarcity has led to the continued reliance on older GPU architectures with relatively low-resource GPUs, such as Volta, in many research settings.
>
> **Reference**
>
> [1] Strati F, Elvinger P, Kerimoglu T, et al. ML Training with Cloud GPU Shortages: Is Cross-Region the Answer?[C]//Proceedings of the 4th Workshop on Machine Learning and Systems. 2024: 107-116.
>
> [2] Sparkes M. AI developers feel chip squeeze[J]. 2023.
>
> [3] Luu H, Pumperla M, Zhang Z. The Future of MLOps[M]//MLOps with Ray: Best Practices and Strategies for Adopting Machine Learning Operations. Berkeley, CA: Apress, 2024: 305-327.
>
> [4] Kristensen J, Wender D, Anthony C. Commodification of compute[J]. arXiv preprint arXiv:2406.19261, 2024.
>
> [5] Guido Appenzeller, Matt Bornstein, and Martin Casado. Navigating the high cost of ai compute. Andreessen Horowitz, April 2023.
>
> [6] Josh Constine and Veronica Mercado. The ai compute shortage explained by nvidia, crusoe, & mosaicml. SignalFire Blog, August 2023.

---

> ### Author Response · Authors · 2024-11-25
> **Hope for the feedback**
>
> Dear Reviewer XLQ8,
>
> Thanks for your valuable time and insightful comments. We deeply appreciate your constructive feedback and hope that our revisions have adequately addressed the concerns raised in your initial reviews. We look forward to your insights on the updated manuscript and any additional suggestions you may have for further improvement.
>
> As the deadline for the Author/Reviewer discussion approaches, please let us know if you require any additional information or clarification. We are fully committed to refining our work and are eager to engage in further discussions to enhance the quality of the submission. Thank you once again for your consideration and guidance!

---

> ### Author Response · Authors · 2024-11-27
>
> Dear Reviewer XLQ8,
>
> Thanks for recognizing the value of our work. Based on the proposed optimization strategies, we here summarize some high-level and general insights that can be applied to a broader range:
>  - **Pipeline**. As discussed above, FastAttention leverages pipeline optimization by overlapping GEMM operations with element-wise computations (e.g., Softmax). This optimization can also be extended to modern hardware architectures that support concurrent execution of matrix units and vector units. For instance, the Hopper architecture (the new generation CUDA architecture) introduces warpgroup-wide WGMMA instructions, enabling overlap between GEMM operations (executed on Tensor Cores) and non-GEMM operations (executed on CUDA Cores) [1]. Similarly, TPU v4 supports a pipeline programming model, where the Vector Processing Units (VPUs) and Matrix Multiply Units (MXUs) can execute computations concurrently [2,3]. The Intel Gaudi 2 architecture also supports such optimizations, facilitating concurrent execution of different computational tasks [4].
>  - **Two-level tiling and Row-wise Partitioning** can be applied for the architectures that the Cube unit and Vector unit have indenpendent buffer sizes, e.g., Ascend 910B series.
>  - **Tiling-mask**. Our tiling-mask strategy is applicable for the architectures with SIMD programming model, such as AMD GPUs [5]. In the SIMT (Single Instruction, Multiple Threads) model, causal masking is determined solely by the relative positions in the sequence for each attention score. However, implementing this approach in the SIMD (Single Instruction, Multiple Data) model, such as through a `for` loop, can be highly inefficient.
>  - **Tiling-AllReduce** can be applied to architectures that enable the concurrent execution of AllReduce communication and computation, e.g., all versions of Ascend NPUs. Theoretically, the CUDA architecture also supports this implementation, with [6] providing an example of its potential application.
>  - **Datalayout redesign** is applicable to all the Volta-architecture GPUs.
>  - **CPU-GPU cooperative strategy** can be adapted for all architectures.
>
> [1] NVIDIA. Parallel Thread Execution ISA Version 8.4, 2024.
>
> [2] TPU v4: An Optically Reconfigurable Supercomputer for Machine Learning with Hardware Support for Embeddings.
>
> [3] TPU - University of Illinois Urbana-Champaign.
>
> [4] Intel-Gaudi2-AI-Accelerators-whitepaper.
>
> [5] AMD: amd-gcn1-architecture.
>
> [6] NanoFlow: Towards Optimal Large Language Model Serving Throughput.

---

> ### Author Response · Authors · 2024-11-30
> **Any further suggestions or questions?**
>
> Dear Reviewer XLQ8,
>
> We would like to express our sincere gratitude for the time and effort you have devoted to reviewing our work and engaging in the current discussion period. We understand how demanding this time can be, especially with your own commitments, and truly appreciate the thoughtful attention you have given to our paper.
>
> We are deeply excited about this paper and its findings, and we greatly value the opportunity to engage in meaningful discussions with you. Please feel free to reach out with any questions, and we are happy to provide further clarifications.

---

> ### Author Response · Authors · 2024-12-02
> **Kind Reminder: Last Day of Reviewer-Author Discussion**
>
> Dear Reviewer XLQ8,
>
> Thank you once again for your efforts and thoughtful comments. With only 24 hours remaining in the Author/Reviewer discussion period, we kindly ask if you could review our responses to your concerns and let us know if there are any additional questions or unresolved points. We would be happy to address them promptly.
>
> If you find our responses satisfactory, we would greatly appreciate it if you could consider reflecting this in your final score. Your valuable feedback is instrumental in improving the quality of our work, and we sincerely thank you for your contributions to this process.
>
> Best regards,
>
> The authors of Submission 9395

---

> ### Comment · Reviewer_XLQ8 · 2024-12-02
>
> Raised the score to 6, as this is a very solid system work and can be applied to the industry applications on the target architecture, even though I still feel it is not that novel.

---

> ### Author Response · Authors · 2024-12-03
> **Thanks for your support**
>
> Dear Reviewer XLQ8,
>
> Thank you very much for your support! Your recognition of the improvements in our work is highly appreciated. We are also grateful for the time and effort you have dedicated to reviewing our paper.
>
> Best regards!

---

### Official Review · Reviewer_3aYo · 2024-11-03

**Soundness:** 3
**Presentation:** 2
**Contribution:** 3
**Rating:** 5
**Confidence:** 4

**Summary:**

The paper proposes to refactor the FlashAttention (used only on high-end GPUs A/H-series because of the tensor core architecture) to NPUs and earlier generation Volta (V100) GPUs. The adaptation is poses some challenges due to the underlying hardware architectures (NPUs have AI core and Vector units whereas V100s don't have tensor cores), therefore, the proposed new two-level tiling strategy for memory and commute efficiencies. On Vota GPUs, there is a further enhancement in the form of CPU-GPU based cooperative strategy for better memory usage. The proposed refactorings show efficient improvments in speedup and throughput on the corresponding hardware chips when compared to the vanilla case.

**Strengths:**

- The paper is some what reasonably written, however was able to follow along and understand the presented concepts.
- The strengths of the paper are in the new tiling strategies be it be memory efficiency or the all-reduce communication efficiencies.
- Given the shortage of GPUs and at the outset to better exploit the available resources, FastAttention kind of a technique is very much appreciated and in that sense this approach is highly valuable.

**Weaknesses:**

Please follow the questions section for the detailed weaknesses and the corresponding questions.

**Questions:**

- In section 4.1 there is a description on the how the attention operations work in the proposed tiling strategy. It is not fully clear from the description as to how they work. Please address the following questions on this.
  - Please represent the tiling mathematically with equations showing how the matrices are tiled along with their dimensions
  - A step-by-step breakdown of operations on the tiles, especially in a two-level tiling strategy.
  - In 2-3 sentences, explain how FastAttention tiling strategies compare with that of the existing FlashAttention.
- In section 4.2, there is this new term `Linear calculation` is not defined earlier, nor is easy to decipher. On that note, Please define clearly what these operations are, very important to understand given that there is so much optimization on these operations.
  - Please define "Linear calculation" when it's first introduced and clarify how these calculations take place in the overall attention mechanism, especially when the two stage tiling is in effect.
- what does this sentence mean `Given the CuTe library typically focuses ... ` (around lines 298 and 299). Please provide clarification on this.
- In section 4.3, the data layout is changed with CuTe to support Volta-GPUs. Two questions on this are
  1. Data layout adaption should be done for a new LLM architecture or is this architecture agnostic? There are some details in Appendix B but it is not clear whether the process is manual or can be reproduced by using an algorithm.
  2. Then, besides there are bank conflicts being resolved, is there a procedure that can be followed or is it pretty manual and should be handled with care for each model? Note that there are not many details on how this was achieved either in Appendix or in the main paper.
  - For both the questions, please provide procedural details that helped close the gap in porting the FlashAttention to these older generation GPUs.
- In section 4.4, the terms `L_{CPU}` and `L_{GPU}` are introduced, but never defined. Please add definitions to those terms.
- In Figure 8, please add a comparison with huggingface transformers implementation. This can help show the improvements over the vanilla implementations, especially there is a significant user base.

### minor issues:
- `thread in a Wrap.` at line 287 should be `warp`

---

> ### Author Response · Authors · 2024-11-22
> **Response to Reviewer 3aYo (1/3)**
>
> Dear Reviewer 3aYo,
>
> Thank you very much for your valuable comments. We highly appreciate your positive evaluation and insightful feedback, which provides us with valuable opportunities to improve our manuscript. We address your concerns and questions as follows:
>
> **Question 1:** Please represent the tiling mathematically with equations showing how the matrices are tiled along with their dimensions.
>
> **Reply:** Given the matrices $Q,K,V \in R^{B \times N \times S \times d}$ and $block \; sizes \; B_r$ and $B_c$, the $Q$ matrix will be splited along S dimension into $Q_1, Q_2, ... , Q_r \in R^{B_r \times d}$. In our two-level tiling strategy, the first level adapts the larger block size $B_r$. There are a total of $B \times N \times \lceil \frac{S}{B_r} \rceil$ large blocks and these blocks will be distributed across AI Cores.
> In each AI Core, it will follow the computations as blow:
>
> $Matrices \quad K,V \in R^{S \times d} \quad O_i,Q_i \in R^{B_r \times d} \quad (The \ First \ Level)$ \
> $K,V: block \, size = B_c \quad K_{1},...,K_{c} \, and \, V_{1},...,V_{c}\in R^{B_c \times d}$ \
> $Init: O_{i}^{(0)} \in R^{B_r \times d},l_i^{(0)} \in R^{B_r}, m_i^{(0)} =(-\infty)_{B_r} \in R^{B_r}$
>
> $for \ 1 \leq j \leq c:$ \
> $\qquad  S_i^{(j)} = Q_{i}K_{j}^T \in R^{B_r \times B_c} (Cube)$ \
> $\qquad  S_i^{(j)}: block \ size=B_b \quad S_{i1},...,S_{ib} \in R^{B_b \times B_c} (Second \ Level)$ \
> $\qquad  for \: 1 \leq k \leq b: \qquad (Vector)$ \
> $\qquad \qquad  m_{ik}^{(j)} = max(m_{ik}^{(j-1)},rowmax(S_{ik}^{(j)})) \in R^{B_b}$ \
> $\qquad \qquad  P_{ik}^{(j)} = exp(S_{ik}^{(j)} - m_{ik}^{(j)}) \in R^{B_b \times B_c}$ \
> $\qquad \qquad  l_{ik}^{(j)} = e^{m_{ik}^{(j-1)}-m_{ik}^{(j)}} \, l_{ik}^{(j-1)} +rowsum(P_{ik}^{(j)}) \in R^{B_b}$ \
> $\qquad M_{i}^{(j)} = P_{i}^{(j)}V_{j} \in R^{B_r \times d}\quad(Cube)$ \
> $\qquad O_{i}^{(j)} = diag(e^{m_{i}^{(j-1)}-m_{i}^{(j)}})^{-1}O_{i}^{(j-1)} + M_{i}^j \quad(Vector)$ \
> $O_{i} = diag(l_i^c)^{-1}O_{i}^c \quad (Vector)$
>
> In the tiling-AllReduce strategy, once the attention for $𝑁$ heads in a sequence is completed, the large blocks proceed to perform the Linear and AllReduce operations:
>
> $O_i = attention(Q_i,K,V)\in R^{B_r \times Nd}, W_o \in R^{Nd \times H}$ \
> $If \ this \ is \ the \ last \ block \ for \ the \ current \ sequence:$ \
> $\qquad Linear\\_out = OW_o \in R^{B_r \times H}$ \
> $\qquad Final\\_Out = Allreduce(Linear\\_out)$
>
> Thank you agian for your constructive comments.
>
> **Question 2:** A step-by-step breakdown of operations on the tiles, especially in a two-level tiling strategy.
>
> **Reply:** We provide a more detailed diagram in our revised version to illustrate the proposed two-level tiling strategy. And it can also be found in the **Figure 1** of the official comment provided below. Specifically, the $Q \in R^{B\times N\times S\times d}$ matrix is divided into multiple blocks with large block size along the S-dimension. These blocks are distributed across AI Cores.
>
> Within each AI Core, the process proceeds as follows::
> - The Cube unit computes the matrix multiplication $S_0 = Q_0K^T$ and stores the results in GM. During the computation of the $Q_0$ block, a double-buffering technique is employed to simultaneously load the $Q_1$ block, effectively eliminating the I/O overhead from GM.
> - The Cube unit then starts the next matrix multiplication $S_1 = Q_1K^T$.
> - Simultaneously, the Vector unit divides $S_0$ matrix into multiple small blocks $S_{01},...,S_{0b}$ along the row dimension. The Vector unit also utilizes the double-buffering technique and loads the small block $S_{0i}$ at a time and performs the $Softmax$ computation.
> - Once the Vector unit completes the $Softmax$ computation for $S_0$, the result matrix $P_0$ is stored in GM.
> - The Cube unit then completes the computation for the $Q_1$ block and loads $P_0$ from GM to calculate $P_0*V$. Meanwhile, the Vector unit begins the $Softmax$ computation for $S_1$.
> - The Cube unit computes the $O_0=P_0V$ and stores $O_0$ in GM. Concurrently, the Vector unit completes the $Softmax$ computation for $S_1$. Following this, the Cube unit calculates $P_1V$ while the Vector unit updates the $O_0$ results in parallel.
> - Once the AI Core completes the computation for the $Q_0$ block, it proceeds to load the next $Q_i$ block, repeating this procedure until all $Q_i$ blocks have been fully processed.
>
> [Figure 1](https://anonymous.4open.science/r/iclr2025-rebuttal/two-level-tiling.pdf). The two-level tiling strategy that employs the larger block size for Cube unit and maintains the smaller block size for Vector unit.

---

> ### Author Response · Authors · 2024-11-22
> **Response to Reviewer 3aYo (2/3)**
>
> **Question 3** In 2-3 sentences, explain how FastAttention tiling strategies compare with that of the existing FlashAttention.
>
> **Reply:** Thank you. We mark four level of siginificant differences in tiling strategies between FlashAttention and our FastAttention:
> 1. **Pipeline level.** Due to the benefits of decoupled architectures of NPUs, our strategy features a fine-grained pipeline between the Cube unit and the Vector unit, which **naturally allows the efficient overlap of the softmax computation with the GEMM (General Matrix Multiplication)**. In contrast, FlashAttention series are not conveniently supportive to this kind of pipelines.
> 2. **Cache level.** Our two-level tiling strategy encourages the assignment of a larger block size to the Cube unit to fully leverage its computational power, while a smaller block size should be assigned to the Vector unit to accommodate the varying L1 buffer sizes and reduce synchronization overhead between the Cube and Vector units. In contrast, FlashAttention employs a small block size for both Tensor Cores and CUDA Cores.
> 3. **Computation level.** Moreover, our FastAttention can split the matrix along the row dimension for the Vector unit to reduce the number of updates (e.g., rowmax and the P matrix) during the softmax computation, whereas FlashAttention does not.
> 4. **Communication level.** Furthermore, the tiling-AllReduce strategy overlaps the computation with AllReduce communication to reduce communication overhead, which is a feature lacking in the FlashAttention series.
>
> **Question 4:** In section 4.2, there is this new term Linear calculation is not defined earlier, nor is easy to decipher. On that note, Please define clearly what these operations are, very important to understand given that there is so much optimization on these operations.
> Please define "Linear calculation" when it's first introduced and clarify how these calculations take place in the overall attention mechanism, especially when the two stage tiling is in effect.
>
> **Reply:** Thank you for your kind reminder. For the self-attention calculation, $$x_{out} = Softmax(\frac{QK^T}{\sqrt{d}})·V·W_o + x$$
> FlashAttention series accelerates the calculation of $x_o=Softmax(\frac{QK^T}{\sqrt{d}})·V$, the Linear calculation implys the multiplication of $x_o$ and the $W_o$ matrix.
>
> **Question 5:** what does this sentence mean Given the CuTe library typically focuses ... (around lines 298 and 299). Please provide clarification on this.
>
> **Reply:** Thanks. We would like to deeply elaborate on the meaning involved in the sentence. For correct and fast matrix multiplication using the CuTe library, it is necessary to define global and shared memory layouts and copy atoms. In FlashAttention2, these traits are defined in `kernel_traits.h`, such as `SmemLayoutQ` and `GmemLayoutAtom`. Unfortunately, there are no examples in the Cutlass/CuTe library showing how to correctly define them for Volta architecture.
>
> For instance, in the file `test\unit\gemm\device\default_gemm_configuration.hpp`, structs defined for Ampere MMA arguments contain a shared memory layout and a copy atom:
> ```
> template <typename Element, typename Layout, int Alignment, int SizeK>
> struct DefaultGemm_TensorOpSm80_OperandA;
>
> template <typename Element, typename Layout, int Alignment, int SizeK>
> struct DefaultGemm_TensorOpSm80_OperandB;
> ```
> The file also contains the struct definition with MMA parameters:
> ```
> template <typename LayoutA, typename LayoutB, typename LayoutC>
> struct DefaultGemmConfigurationToCutlass3Types<
>     arch::OpClassTensorOp, arch::Sm80,
>     half_t, LayoutA,
>     half_t, LayoutB,
>     float, LayoutC,
>     float>
> ```
> Moreover, there are many unit tests, e.g., `sm80_gemm_f16_f16_f32_tensor_op_f32.cu`, use the structs, but there is no examples for Volta architecture.
>
> **Question 6** Data layout adaption should be done for a new LLM architecture or is this architecture agnostic? There are some details in Appendix B but it is not clear whether the process is manual or can be reproduced by using an algorithm.
>
> **Reply:** Our data layout design is LLM architecture-agnostic, which is totally consistent with the role of the original FlashAttention2. Notably, the head dimension of attention blocks in a model is the only hyperparameter that affects the memory layout. In our design, similar to FlashAttention2, the head dimension (kHeadDim) is treated as a template parameter, along with other parameters kBlockM, kBlockN, and  kNWarps, to define the `Flash_fwd_kernel_traits` structure.

---

> ### Author Response · Authors · 2024-11-22
> **Response to Reviewer 3aYo (3/3)**
>
> **Question 7:** Then, besides there are bank conflicts being resolved, is there a procedure that can be followed or is it pretty manual and should be handled with care for each model? Note that there are not many details on how this was achieved either in Appendix or in the main paper.
>
> **Reply:** Bank conflicts are the inefficient when tackled using shared memory. In terms of CuTe library, it is essential to employ the correct layout, copy algorithm, and copy atom operations to eliminate these conflicts. To address this challenge, we design an approach that executes two consecutive Volta's MMA operations using only registers without storing intermediate results and redesign the data layout to solve the challenges.
> Similar to FlashAttention2, our method does not require additional procedures to be performed for each model. Its practical usage closely mirrors that of FlashAttention. For example:
> ```
> //FlashAttention2
> from flash_attn import flash_attn_func
> output = flash_attn_func(Q,K,V)
> //VERSUS FastAttention
> from fastattention import fast_attn_func
> output = fast_attn_func(Q,K,V)
> ```
>
> **Question 8:** In section 4.4, the terms L_{CPU} and L_{GPU} are introduced, but never defined. Please add definitions to those terms.
>
> **Reply:** $L_{CPU}$ represents the number of layers where the KV cache is stored on CPUs, while $L_{GPU}$ indicates the number of layers where the KV cache is stored on GPUs.
>
> **Question 9:** In Figure 8, please add a comparison with huggingface transformers implementation. This can help show the improvements over the vanilla implementations, especially there is a significant user base.
>
> **Reply:** Actually, the vanilla implementations in huggingface transformers requires $O(N^2)$ memory complexity for each query. In our experimental setup, the vanilla implementations can only support a sequence length of 2k and 4K. In comparison, with casual mask, our FastAttention achieves speedups of 6.4$\times$ and 8.2$\times$, respectively. Without casual mask, FastAttention yields speedups of 1.94$\times$ and 2.22$\times$,respectively. The detailed experimental results can be found in Figure 8 of our revised version or in the **Figure 2** of the official comment provided below.
>
> [Figure 2](https://anonymous.4open.science/r/iclr2025-rebuttal/fa-v100.pdf).  Performance comparison of FastAttention and xformers' FlashAttention with batch size 8, head size 64, and number heads 32 during the *prefill* stage on a V100.
>
> **Question 10:** minor issues:thread in a Wrap. at line 287 should be warp
>
> **Reply:** Thank you. We corrected this error in our revised version.

---

> ### Author Response · Authors · 2024-11-25
> **Hope for the feedback**
>
> Dear Reviewer 3aYo,
>
> Thanks for your valuable time and insightful comments. We greatly value your constructive feedback and hope that our revisions have addressed the concerns raised in your initial reviews. We eagerly anticipate your thoughts any further suggestions you may have to refine our manuscript.
>
> As the deadline for the Author/Reviewer discussion is approaching, please feel free to let us know if additional clarifications or further details are required from our side. We remain committed to refining our work and are more than willing to engage in further discussions to strengthen the submission. Thank you once again for your consideration and guidance!

---

> ### Author Response · Authors · 2024-11-27
> **Looking forward to your reply**
>
> Dear Reviewer 3aYo,
>
> Thank you again for your comments. Your opinion is highly valued, and we have been committed to providing comprehensive responses. We sincerely hope our efforts to address your concerns. We are delighted to provide any additional data, explanations, or results to further address your concerns at any time. We look forward to your feedback and hope for a positive outcome. Thank you very much for your time and consideration.

---

> ### Author Response · Authors · 2024-11-28
> **Reminder for Feedback**
>
> Dear Reviewer 3aYo,
>
> As the deadline for submitting the revised PDF is only a few hours away, it may not be feasible to incorporate further changes into the current version. We apologize for this constraint at the final stage. However, we are fully committed to addressing any additional questions or concerns leading up to December 3rd.
>
> Here are some important deadlines to keep in mind:
>
> November 27th, 11:59 PM AoE: Last day for authors to upload a revised PDF. After this deadline, no further updates to the manuscript will be possible, and authors will only be able to respond to comments on the forum. If you’d like any changes reflected in the revised manuscript, please inform us before this time.
>
> December 2nd: Last day for reviewers to post messages to the authors (six-day extension). This is the final opportunity to share any remaining concerns with us.
>
> December 3rd: Last day for authors to post messages on the forum (six-day extension). After this date, we will no longer be able to respond to any concerns or feedback.
>
> We sincerely thank you once again for your time, effort, and valuable feedback, which have been instrumental in improving our work!

---

> ### Author Response · Authors · 2024-11-30
> **Any further suggestions or questions?**
>
> Dear Reviewer 3aYo,
>
> We would like to express our sincere gratitude for the time and effort you have devoted to reviewing our work. We understand how demanding this time can be, especially with your own commitments, and truly appreciate the thoughtful attention you have given to our paper.
>
> We are deeply excited about this paper and its findings, and we greatly value the opportunity to engage in meaningful discussions with you. Please feel free to reach out with any questions, and we are happy to provide further clarifications.

---

> ### Author Response · Authors · 2024-12-02
> **Kind Reminder: Final Day of Reviewer-Author Discussion**
>
> Dear Reviewer 3aYo,
>
> Thank you once again for your efforts and thoughtful comments. With only 24 hours remaining in the Author/Reviewer discussion period, we kindly ask if you could review our responses to your concerns and let us know if there are any additional questions or unresolved points. We would be happy to address them promptly.
>
> If you find our responses satisfactory, we would greatly appreciate it if you could consider reflecting this in your final score. Your valuable feedback is instrumental in improving the quality of our work, and we sincerely thank you for your contributions to this process.
>
> Best regards,
>
> The authors of Submission 9395

---

> ### Author Response · Authors · 2024-12-03
> **Request for Further Discussion and Feedback**
>
> Dear Reviewer 3aYo,
>
> Thank you once again for your thorough comments and insightful feedback. As the Author/Reviewer discussion period is nearing its conclusion, we sincerely hope to engage in further dialogue with you to address any remaining concerns or questions you may have.
>
>
> We look forward to hearing from you soon.
>
>
> Best regards!

---

### Author Response · Authors · 2024-11-22
**Update of manuscript**

Dear reviewers,

We deeply appreciate your valuable and constructive feedback on our manuscript. In response to your comments and suggestions, we have carefully revised the manuscript and **marked all changes in blue** to facilitate review. Below, we provide a summary of the key updates:

### **Main Text**

**[Section 1]** We provided the appropriate introduction and citation to memory-efficient attention and the work of FlashAttention for AMD GPUs.

**[Section 4.1]** We provided a more detailed diagram to illustrate the proposed two-level tiling strategy. Moreover, we optimized the description of the key innovations and structured them into clear points to enhance readability and clarity.

**[Section 4.2]** We supplemented the definition of "Linear calculation"
and refined the figure captions.

**[Section 4.3]** We clarified the previously vague sentence to enhance precision and readability.

**[Section 4.4]**  We added the definitions for $L_{CPU}$ and $L_{GPU}$.

**[Section 5.2.3]** We have incorporated the experimental results of the Hugging Face Transformers implementation into **Figure 8** and updated the figure to reflect the latest findings.

**[Section 5]** We enhanced the figure captions with the relative relevant details of the experiments.

### **Appendix**

**[Section B]** We supplemented the detailed description of the FastAttention algorithm for NPUs, elaborating on the novel strategies proposed in the manuscript.

---

### Meta-Review · Area_Chair_zrRM · 2024-12-13

**Metareview:**

The paper offers FastAttention, adapting FlashAttention2 to NPUs and low-resource GPUs for efficient inference. While the reviewers applaud the engineering/implementation efforts, there is less consensus on the novelty and research contribution of this work. Therefore, it may not be suited to be accepted at ICLR in its current form.

**Additional Comments On Reviewer Discussion:**

Although not all reviewers show up during the rebuttal process, the contributions of this paper are evaluated with this taken into account.

---

### Decision · Program_Chairs · 2025-01-22

Reject